# From structural polymorphism to structural metamorphosis of the coat protein of flexuous filamentous potato virus Y

Luka Kavčič [1,2], Andreja Kežar [1], Neža Koritnik [1,3], Magda Tušek Žnidarič[4], Tajda Klobučar [1,5], Žiga Vičič [1], Franci Merzel[6], Ellie Holden[7,8], Justin L. P. Benesch [7,8] & Marjetka Podobnik [1✉]

The structural diversity and tunability of the capsid proteins (CPs) of various icosahedral and rod-shaped viruses have been well studied and exploited in the development of smart hybrid nanoparticles. However, the potential of CPs of the wide-spread flexuous filamentous plant viruses remains to be explored. Here, we show that we can control the shape, size, RNA encapsidation ability, symmetry, stability and surface functionalization of nanoparticles through structure-based design of CP from potato virus Y (PVY). We provide high-resolution insight into CP-based self-assemblies, ranging from large polymorphic or monomorphic filaments to smaller annular, cubic or spherical particles. Furthermore, we show that we can prevent CP self-assembly in bacteria by fusion with a cleavable protein, enabling controlled nanoparticle formation in vitro. Understanding the remarkable structural diversity of PVY CP not only provides possibilities for the production of biodegradable nanoparticles, but may also advance future studies of CP's polymorphism in a biological context.

[1] Department of Molecular Biology and Nanobiotechnology, National Institute of Chemistry, Ljubljana, Slovenia. [2] PhD Program 'Chemical Sciences', Faculty of Chemistry and Chemical Technology, University of Ljubljana, Ljubljana, Slovenia. [3] PhD Program 'Biomedicine', Faculty of Medicine, University of Ljubljana, Ljubljana, Slovenia. [4] Department of Biotechnology and Systems Biology, National Institute of Biology, Ljubljana, Slovenia. [5] PhD Program 'Biosciences', Biotechnical Faculty, University of Ljubljana, Ljubljana, Slovenia. [6] Theory Department, National Institute of Chemistry, Ljubljana, Slovenia. [7] Department of Chemistry, University of Oxford, Oxford, UK. [8] Kavli Institute for Nanoscience Discovery, University of Oxford, Oxford, UK. ✉email: marjetka.podobnik@ki.si

Single-stranded RNA (ssRNA) viruses account for nearly half of all plant viruses. Most of them have only one type of structural protein, a capsid (or coat) protein (CP), and form either rod-shaped or flexuous filamentous virions[1]. The latter are much more common[2], with viruses of the genus *Potyvirus* (family *Potyviridae*) representing the largest group[3]. Potyviruses have major economic impact and are responsible for more than half of the world's viral crop damage[4]. Their genomic positive-sense ssRNA of about 10 kb generally encodes ten proteins[3], which includes also the CP, whose copies form a flexuous filamentous capsid with left-handed helical symmetry around the viral ssRNA, as shown by the cryo-EM structures of watermelon mosaic virus (WMV), potato virus Y (PVY), and turnip mosaic virus (TuMV)[5–7]. CP consists of a highly conserved globular core flanked by two large extended regions with a high frequency of structural disorder (Fig. 1a)[5–8]. The C-terminal high intrinsic disorder region (C-IDR) is partially conserved and packaged in the lumen of the virion, supported by the helical ssRNA scaffold. The N-terminal IDR (N-IDR) exhibits very low amino acid conservation. It is exposed on the outer surface of the virion and is critical for the flexible nature of virions, connecting the CP units longitudinally and perpendicular to the filament axis[5–7]. This is in contrast to viruses from the genera of rigid rod-shaped tobamoviruses or hordeiviruses, where the relatively short C- and N-terminal structural elements are exposed on the outer surface of CP, and the connection between the CP units is established by the wedge-shaped CP cores[9,10].

Potyviral CP plays a role in virtually every step of the viral infection cycle, from transmission of the virion by aphids, virus assembly and disassembly, regulation of genome amplification, protein translation, to cell-to-cell and long-distance movement[11]. The structural context in which CP acts during the different phases of the viral cycle is not yet known, however, the intrinsic structural plasticity of CP[5–8] mainly contributed by both IDRs seems to play the crucial role[6,8].

The presence of different structural states or pleomorphism of viral capsids is quite common in enveloped[12,13], icosahedral[14] or even some helical viruses[10,15,16] and has been associated with certain stages of their life cycle[17,18]. In addition, structural polymorphism is a well-known feature of recombinantly produced virus-like particles (VLPs) derived from icosahedral or rod-shaped viruses[14–16]. CP and its mode of self-assembly can be modulated using structural synthetic virology approaches, resulting in symmetric nanoparticles of different shapes and sizes with specific material properties that have great potential for medical, biotechnological, or smart material applications[14,19–21].

Although the structures of several flexuous filamentous potexviruses[22–26] and potyviruses[5–7] have been recently determined, information on the structural diversity of these viruses and their VLPs is lacking[6,7,23,26,27]. In our work, we have investigated the structural landscape of self-assemblies formed by potyviral CP. While the structural analysis of natural supramolecular complexes formed by CPs during the viral life cycle is challenging due to the very complex and dynamic natural context, the successful production of recombinant potyviral VLPs has been reported for different expression systems, preferring plants or bacteria[28]. Interestingly, the cryo-EM structure of PVY VLPs prepared from bacteria, determined at 4.1 Å resolution, showed a markedly different architecture of VLP filaments than the structure of PVY virions, as they consisted of stacked octameric CP rings and did not contain RNA[6]. On the other hand, the structure of TuMV VLPs produced by transient expression in tobacco at 8.0 Å resolution indicated an RNA-free filamentous arrangement of CP units in left-handed helical symmetry[7]. Interestingly, the structure of VLPs determined at 2.6 Å resolution based on CP of another potyvirus, sweet potato feathery mottle virus (SPFMV),

and produced in tobacco by transient expression in the presence of a replicating RNA, showed a virus-like architecture, with ssRNA directing the helical arrangement of CPs along the filament[27]. These studies showed that a specific potyviral CP self-assembles into filaments of a single architectural type under selected experimental conditions, but this differed among the three experimental arrangements, suggesting that polymorphism may also exist within a species of CP under certain conditions. In this study, we found that the wild type PVY CP can indeed form three architecturally distinct types of VLPs simultaneously. Furthermore, through structure-based engineering of PVY CP, we discovered that we can control the formation of a wide range of highly ordered supramolecular assemblies, their architecture, RNA encapsidation, and molecular properties. These can range from various filamentous to ring-shaped, cubic or spherical assemblies with high symmetry, most of which form without a template. To avoid spontaneous CP self-assembly in a complex bacterial environment, we have developed a system for CP self-assembly in vitro that allows the controlled formation of nanoparticles with desired properties. This remarkable structural diversity of PVY CP nanoparticles makes them great candidates for nanobiotechnological applications. Moreover, the high-resolution details about the structural plasticity of PVY CP could pave the way for a better understanding of CP polymorphism in a biological context.

## Results

### Recombinant PVY CP self-assembles into three architecturally distinct types of VLPs.

To investigate the potential of PVY CP to form polymorphic assemblies, we produced PVY VLPs in bacteria. A comprehensive analysis of the cryo-EM data revealed two new filament architectures (Fig. 1b, c; Table 1; Supplementary Figs. 1a, 2) in addition to the predominant RNA-free stacked ring assembly (VLP$^r$) that we had previously observed[6]. 25% of picked particles exhibited left-handed helical symmetry with no RNA packed inside (VLP$^h$), similar to TuMV VLPs[7]. The remaining 8% of picked particles also exhibited left-handed helical symmetry and encapsidated RNA (VLP$^{h+RNA}$), closely resembling PVY virion[6] (Fig. 1c; Supplementary Table 1) and SPFMV VLPs[27]. An improved data analysis procedure seemed to play a crucial role here, as such a distribution of polymorphic filaments was also obtained by reprocessing our previous data[6] (Supplementary Fig. 1b).

The C-IDR is structurally defined only in VLP$^{h+RNA}$ filaments, i.e. in CP$^{h+RNA}$, where the helical scaffold of RNA supports the cone-like organization of C-IDRs in the lumen of the filament (Fig. 1c, d). In the absence of RNA in VLP$^r$ and VLP$^h$, the C-IDR in CP$^r$ and CP$^h$ is disordered, with no traceable cryo-EM density beyond A222 (Supplementary Fig. 2a). The fold of the CP core domain is conserved between the polymorphic filaments, except for the conserved RNA-binding loop S125-G130[6], which adopts different conformations in the presence or absence of RNA (Fig. 1d). In all three structures, there was no cryo-EM density for the first 41 residues of N-IDR, exposed on the outer surface of the filaments (Supplementary Fig. 2a). Beyond H42, N-IDR in CP$^{h+RNA}$ folds similarly to that in PVY virus (Supplementary Fig. 2b), whereas N-IDR in CP$^r$ and CP$^h$ takes a different turn at K53 (Fig. 1d). Thus, the structural plasticity of PVY CP, in particular the two IDRs and the conserved RNA-binding loop S125-G130, enables the polymorphism of PVY VLPs produced in bacteria. In the absence of ssRNA, the N-IDRs adopt two slightly different conformations, as seen in CP$^r$ and CP$^h$ (Supplementary Fig. 2c) allowing the formation of two different types of RNA-free filaments (Supplementary Fig. 2d), with the stacked octameric ring assembly being the more stable and therefore predominant form.

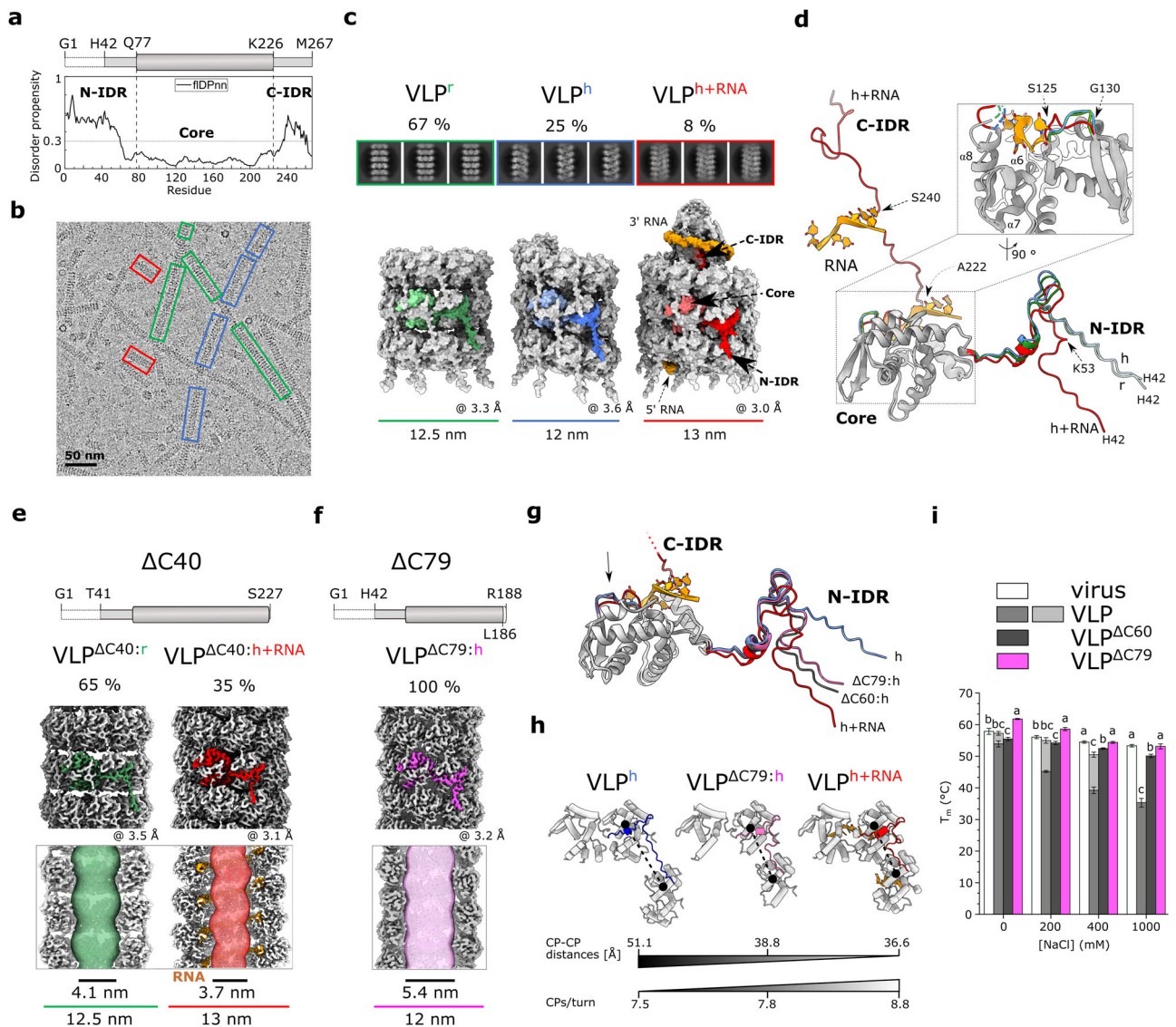

**Fig. 1 Structural polymorphism of recombinant PVY CP. a** Top: schematic representation of CP, the marked residues delineate N-IDR, Core and C-IDR. Residues G1-T41 are not resolved in the cryo-EM maps. Bottom: flDPnn[96] prediction of structural disorder in CP (threshold = 0.3). **b** Cryo-EM micrograph of wild type VLPs. Architecturally distinct filament types are marked: red: VLP[h+RNA], blue: VLP[h-RNA], green: VLP[r]. **c** Cryo-EM 2D class averages and 3D reconstructions of the three VLP types. A CP subunit within each VLP is colored, color code as in **b**. Orange: RNA. The percentage of each particle type is indicated above the 2D classes. The overall resolution (in Å) and diameter (in nm) of the filaments are indicated. **d** Superposition of CP subunits of the three VLP forms. IDRs, the S125-G130 RNA-binding loop, and RNA (sticks) are colored as in **b**. For RMSD values, see Supplementary Fig. 2c. Top: schematic representation of CP[ΔC40] (**e**) and CP[ΔC79] (**f**). Middle: cryo-EM 3D reconstructions of the corresponding VLPs with subunits highlighted in red (VLP[ΔC40:h+RNA]) or green (VLP[ΔC40:r]) (**e**) and pink (VLP[ΔC79:h]) (**f**). Bottom: cross-section of filaments with corresponding filament widths and inner channel diameters (MOLE 2.5[97]). **g** Superposition of atomic models of different CPs showing the N-IDRs of CP[ΔC60:h] (black), CP[ΔC79:h] (pink), CP[h] (blue) and CP[h+RNA] (red). As the focus is on different N-IDR conformations, the CP[h+RNA] C-IDR is not fully shown; its direction is indicated by a dotted line. The black arrow marks the RNA-binding loop. For RMSD values, see Supplementary Fig. 4c. **h** Packing of CP subunits connected by N-IDR in VLP[h] (blue), VLP[ΔC79:h] (pink), and VLP[h+RNA] (red). CP-CP distances between N113-Cα atoms of the subunits in adjacent helical turns are shown and the number of CP units per helix turn. **i** Thermal stability. Melting temperatures ($T_m$) are shown as mean ± SD ($N = 6$) for VLP[ΔC60] (dark gray), VLP[ΔC79] (pink), VLP (gray/light gray), and PVY virus (white) at different NaCl concentrations at pH 7.0. Statistical significance was determined by one-way ANOVA with Tukey's multiple comparison test ($p = 0.001$). Markers (**a–c**) indicate a statistically significant difference. The source data for panels **a** and **i** are provided in the Supplementary Data file.

**C-terminal truncation of CP reduces the architectural diversity of VLPs.** Potyviral CP with deleted C-IDR still forms filaments[6,29]. Because C-IDR is structurally defined in wild type VLPs only in the presence of ssRNA (VLP[h+RNA]), we examined how the absence of C-IDR affects filament architecture.

We prepared VLPs consisting of the CP units lacking 40 C-terminal residues, i.e whole C-IDR (CP[ΔC40]), and analyzed them by cryo-EM (Fig. 1e). Interestingly, we detected only two

different architectures of filaments. 65% of them had the stacked-ring architecture (VLP[ΔC40:r]) and 35% had left-handed helical symmetry and encapsidated RNA (VLP[ΔC40:h+RNA])(Fig. 1e; Supplementary Fig. 3). The structures of the CP[ΔC40:r] and CP[ΔC40:h+RNA] subunits and the helical parameters of their VLPs were comparable to those of their wild type counterparts (Supplementary Fig. 4a; Table 1). This indicates that luminal C-IDR is not essential either for filament formation or RNA

**Table 1 Cryo-EM data collection, refinement and validation statistics.**

| | VLPr | VLPh | VLPh +RNA | VLPΔC40:r | VLPΔC40:h +RNA | VLPΔC60:h | VLPΔC79:h | trCP (H2Tdouble ring) | trCPK176C (cubes, global) | trCPK176C (cubes, local) | VLPT43C +D136C:h+RNA |
|---|---|---|---|---|---|---|---|---|---|---|---|
| | (EMD-17046) | (EMD-17047) | (EMD-17048) | (EMD-17049) | (EMD-17050) | (EMD-17051) | (EMD-17052) | (EMD-17053) | (EMD-17062) | (EMD-17063) | (EMD-17072) |
| | (PDB 8OPA) | (PDB 8OPB) | (PDB 8OPC) | (PDB 8OPD) | (PDB 8OPE) | (PDB 8OPF) | (PDB 8OPG) | (PDB 8OPH) | (PDB 8OPJ) | (PDB 8OPK) | (PDB 8OPL) |
| Percentage of particles detected in the sample (%) | 67 | 25 | 8 | 65 | 35 | 100 | 100 | 93 | 100 | | 100 |
| **Data collection and processing** | | | | | | | | | | | |
| Magnification | ×150,000 | | | ×75,000 | | ×150,000 | ×150,000 | ×150,000 | ×165,000 | | ×150,000 |
| Voltage (kV) | 200 | | | 300 | | 200 | 200 | 200 | 300 | | 200 |
| Microscope | Glacios | | | Titan Krios | | Glacios | Glacios | Glacios | Titan Krios | | Glacios |
| Electron exposure (e- Å$^{-2}$) | 40 | | | 84 | | 40 | 42 | 43 | 32 | | 40 |
| Defocus range (μm) | −0.8 to −2.1 (step 0.3) | | | −1.3 and −0.4 (step 0.3) | | −0.8 to −2.1 (step 0.3) | −0.8 to −2.1 (step 0.3) | −0.8 to −2.1 (step 0.3) | −0.3 to −3.6 (step 0.3) | −0.3 to −3.6 (step 0.3) | −0.8 to −2.1 (step 0.3) |
| Pixel size (Å) | 0.950 | | | 1.063 | | 0.950 | 0.950 | 0.950 | 0.822 | | 0.950 |
| Symmetry imposed | helical, C8 | helical, C1 | helical, C1 | helical, C8 | helical, C1 | helical, C1 | helical, C1 | C8 | O | C1 | helical, C1 |
| Helical rise (Å) | 43.24 | 5.88 | 3.97 | 43.29 | 4.05 | 4.65 | 4.70 | 39.30 | N/A | N/A | 4.01 |
| Helical twist (°) | 13.78 | −48.11 | −40.96 | 15.04 | −41.05 | −46.36 | −46.47 | −3.59 | N/A | N/A | −41.00 |
| Total movies | 502 | | | 4805 | | 2020 | 491 | 2862 | 9480 | | 461 |
| EMPIAR-ID | EMPIAR-11545 | | | EMPIAR-11546 | | / | EMPIAR-11547 | EMPIAR-11548 | EMPIAR-11549 | | EMPIAR-11550 |
| Initial particle/segment (no.) | 109,796 | 27,500 | 10,266 | 2,966,850 | 152,296 | 788,376 | 102,608 | 1,406,546 | 1,580,128 | | 195,881 |
| Final particle/segment (no.) | 49,140 | 27,500 | 10,266 | 399,617 | 152,296 | 222,692 | 66,934 | 174,238 | 210,779 | 843,116[a] | 141,112 |
| Map resolution (Å) | 3.3 | 3.6 | 3.0 | 3.5 | 3.1 | 2.8 | 3.2 | 2.9 | 3.0 | 3.2 | 2.4 |
| FSC threshold | 0.143 | 0.143 | 0.143 | 0.143 | 0.143 | 0.143 | 0.143 | 0.143 | 0.143 | 0.143 | 0.143 |
| **Refinement** | | | | | | | | | | | |
| Initial model used (PDB-ID) | 6HXZ | 6HXX | 6HXX | 6HXZ | 6HXX | 6HXX | 6HXX | 8OPD | 8OPK | 8OPD | 6HXX |
| Model resolution (Å) | 3.5 | 3.8 | 3.1 | 3.6 | 3.1 | 2.9 | 3.4 | 3.1 | 3.5 | 3.2 | 2.6 |
| FSC threshold | 0.5 | 0.5 | 0.5 | 0.5 | 0.5 | 0.5 | 0.5 | 0.5 | 0.5 | 0.5 | 0.5 |
| **Model composition** | | | | | | | | | | | |
| Number of protein chains | 24 | 16 | 28 | 24 | 21 | 19 | 19 | 16 | 48 | 16 | 27 |
| Number of RNA chains | 0 | 0 | 28 | 0 | 21 | 0 | 0 | 0 | 0 | 0 | 27 |
| Non-hydrogen atoms | 34,992 | 23,328 | 53,088 | 66,672 | 33,411 | 22,401 | 22,059 | 21,824 | 65,352 | 21,784 | 50,679 |
| Protein residues | 4344 | 2896 | 6328 | 4344 | 3906 | 2793 | 2755 | 2704 | 8112 | 2704 | 6048 |
| Nucleotide residues | 0 | 0 | 140 | 0 | 105 | 0 | 0 | 0 | 0 | 0 | 135 |
| Ligands | 0 | 0 | 0 | 0 | 0 | 0 | 0 | 0 | 0 | 0 | 0 |
| **R.m.s. deviations** | | | | | | | | | | | |
| Bond lengths (Å) | 0.006 | 0.005 | 0.005 | 0.008 | 0.006 | 0.005 | 0.005 | 0.005 | 0.007 | 0.007 | 0.005 |
| Bond angles (°) | 1.175 | 1.157 | 1.033 | 1.215 | 0.969 | 0.991 | 1.048 | 1.033 | 1.148 | 1.147 | 0.969 |
| **Validation** | | | | | | | | | | | |
| MolProbity score | 1.76 | 1.87 | 1.87 | 1.99 | 1.31 | 1.47 | 1.70 | 1.72 | 1.73 | 1.70 | 1.20 |
| Clashscore | 7.55 | 9.17 | 8.83 | 11.53 | 5.64 | 7.88 | 7.48 | 6.64 | 7.12 | 6.64 | 4.21 |
| Rotamers outliers (%) | 0.00 | 0.63 | 1.08 | 0.03 | 0.24 | 0.53 | 0.78 | 0.60 | 0.72 | 0.68 | 0.71 |
| Ramachandran plot | | | | | | | | | | | |

**Table 1 (continued)**

| | VLPr | VLPh | VLPh+RNA | VLPΔC40:r | VLPΔC40:h+RNA | VLPΔC60:h | VLPΔC79:h | trCP (H2T double ring) | trCPK176C (cubes, global) | trCPK176C (cubes, local) | VLPT43C +D136Ch+RNA |
|---|---|---|---|---|---|---|---|---|---|---|---|
| | (EMD-17046) | (EMD-17047) | (EMD-17048) | (EMD-17049) | (EMD-17050) | (EMD-17051) | (EMD-17052) | (EMD-17053) | (EMD-17062) | (EMD-17063) | (EMD-17072) |
| | (PDB 8OPA) | (PDB 8OPB) | (PDB 8OPC) | (PDB 8OPD) | (PDB 8OPE) | (PDB 8OPF) | (PDB 8OPG) | (PDB 8OPH) | (PDB 8OPJ) | (PDB 8OPK) | (PDB 8OPL) |
| Favored (%) | 94.97 | 94.41 | 94.64 | 93.85 | 98.91 | 97.93 | 95.80 | 94.87 | 95.21 | 95.21 | 99.55 |
| Allowed (%) | 5.03 | 5.03 | 5.36 | 6.15 | 1.09 | 2.07 | 4.20 | 5.13 | 4.79 | 4.79 | 0.45 |
| Disallowed (%) | 0.00 | 0.63 | 0.00 | 0.00 | 0.00 | 0.00 | 0.00 | 0.00 | 0.00 | 0.00 | 0.00 |
| Correlation coefficient (CC) | 0.84 | 0.83 | 0.87 | 0.74 | 0.87 | 0.84 | 0.83 | 0.78 | 0.77 | 0.72 | 0.86 |

'N/A'...not applicable for non-helical reconstructions.
aDesignates the number of particle images after C4 symmetry expansion.

encapsidation. The absence of C-IDR facilitates the accessibility of the RNA-binding site, resulting in a significantly increased proportion of VLPΔC40:h+RNA filaments. VLPΔC40:r and VLPΔC40:h+RNA filaments are long flexible and hollow nanotubes with an inner dimeter of 4.1 nm and 3.7 nm, respectively (Fig. 1e; Supplementary Fig. 4b).

To prevent encapsidation of ssRNA, we further truncated the C-terminus, excluding 60 C-terminal residues (CPΔC60) containing both the C-IDR and the α8-helix, which is placed opposite to the RNA-binding loop and forms part of the RNA-binding cleft (Fig. 1d). This indeed led to formation of RNA-free filaments. Interestingly, the filaments had monomorphic helical architecture, VLPΔC60:h, with unique helical parameters (Supplementary Figs. 3, 4c; Table 1). Because we found that 19 C-terminal residues of this construct, including those of the α7-helix (Fig. 1d), were not defined by cryo-EM density, we prepared another deletion mutant, CPΔC79, excluding these residues (Fig. 1f; Supplementary Fig. 3). The VLPΔC79:h filaments had a similar structure to VLPΔC60:h, but a better-defined C-terminal part (Fig. 1g; Supplementary Fig. 4d; Table 1). The disrupted RNA-binding cleft in CPΔC60/CPΔC79 thus lost the ability to bind RNA, and consequently the RNA-binding loop S125-G130 adopted the conformation found in the RNA-free filaments VLPh and in VLPr (Fig. 1g; Supplementary Fig. 4d). On the other hand, the fold of the N-IDR in CPΔC60/CPΔC79 resembled that found in RNA-encapsidating filaments, consistent with the CP-CP distances along the VLPΔC79:h filament being closer to VLPh+RNA than to VLPh (Fig. 1h). Thus, disruption of the RNA-binding cleft by large truncation of the C-terminal part of CP prevents RNA binding and also leads to the formation of monomorphic VLPs. VLPΔC79:h filaments are compact flexible hollow nanotubes with an inner channel diameter of 5.4 nm (Fig. 1f), with thermal stability higher than that of wild type VLPs and comparable to that of PVY virions (Fig. 1i).

**Simultaneous truncation of both CP IDRs leads to formation of stable octameric rings**. The CP N-IDR plays an important role in filament assembly by participating in inter- and intra-ring interactions[6]. Moreover, we have shown here that the structural plasticity of this region enables variability in the packing arrangement of CP subunits in filaments, and thus polymorphism (Fig. 1d; Supplementary Fig. 2c). We have previously shown that truncation of the N-IDR at G40 results in an insoluble protein, whereas CPΔN49 and CPΔN49C40 with both IDRs truncated, self-assemble into single octameric rings and their short stacks[6]. To facilitate sample preparation for further structural analysis, we attached the His6-tag to the C-terminus of CPΔN49C40 (trCP) (Fig. 2a). Cryo-EM revealed that the affinity-purified sample consisted predominantly of trCP double octameric rings assembled in a head-to-tail (H2T) orientation (Fig. 2a; Supplementary Fig. 5). In addition, we observed some shorter filaments (<5%), with RNA-free stacked-ring or helical architecture, with the helical parameters similar to those of VLPΔC79:h (Fig. 2a; Supplementary Fig. 6a).

In wild type VLPr filaments, the N-IDR is responsible for the axial connection of the octameric rings, with no obvious interactions between the core regions of the CPs (Supplementary Fig. 2c)[6]. The truncation of N-IDR reduces the axial separation of the rings by 3.9 Å and shifts the twist angle in H2T double rings compared with VLPr filaments (Fig. 2b; Table 1). The blob of density in the center of the two rings (Fig. 2c), which was already observed in the 2D class averages, was assigned to a cluster of His6-tags, because it was absent in the 2D class averages after the removal of the His6-tag, which also led to the dissociation of double rings into single rings (Fig. 2c).

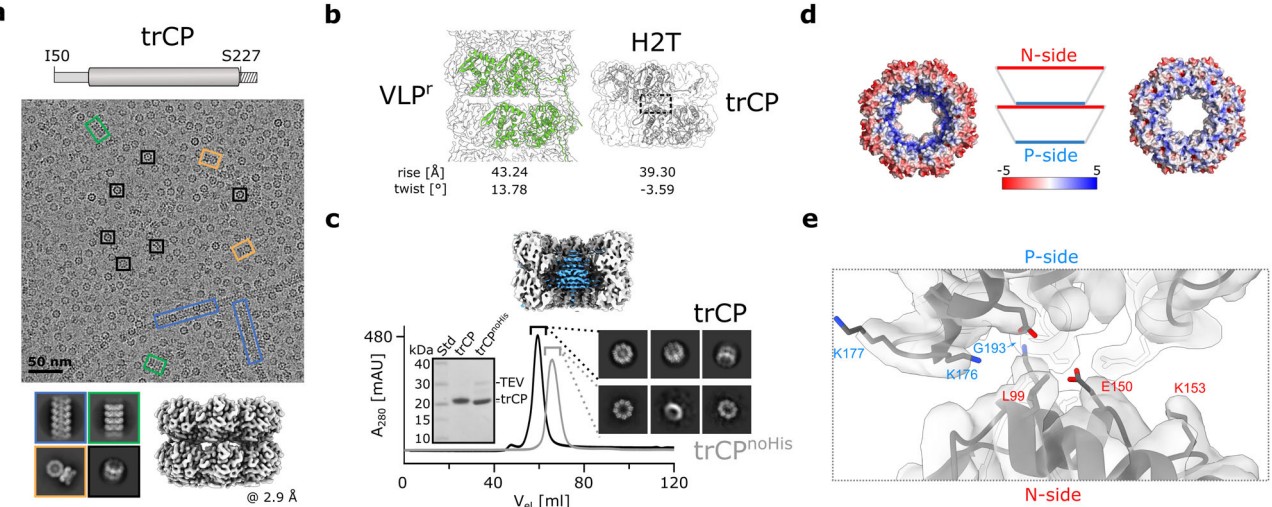

**Fig. 2 CP with truncated IDRs preferentially forms (double) octameric rings. a** Top: schematic representation of the trCP construct. The dashed box beyond S227 marks the presence of the additional linker with the TEV protease cleavage site and the His$_6$-tag. Middle: representative cryo-EM micrograph of the trCP sample. Black rectangles: H2T-double rings; blue rectangles: RNA-free helical filaments; green rectangles: RNA-free stacked-ring filaments; orange rectangles: orthogonal stacking of two H2T double rings. Bottom: left: corresponding 2D class averages. Right: 3D reconstruction of the H2T-double rings, the overall resolution is indicated below in Å. **b** Organization of CP units in the wild type VLP$^r$ (left) and trCP H2T-double ring (right), with helical parameters shown below. The dashed rectangle highlights the contact between two rings in H2T. **c** Top: cross-section of the 3D reconstruction of the trCP H2T double ring with the central untraceable density in blue. Bottom: comparison of SEC (HiLoad Superdex 200 16/600) profiles of trCP before (black) and after removal of the His$_6$-tag (gray, trCP$^{noHis}$) with the corresponding cryo-EM 2D class averages and SDS-PAGE gel. The source data for this panel are provided as Supplementary Data file. **d** Electrostatic surface of trCP H2T double ring with predominant negative (N-side, left) or positive (P-side, right) charge (APBS[86], $-/+$ 5 $k_B$ T e$_c^{-1}$). **e** Model of the trCP H2T double ring in the cryo-EM density map showing the non-conserved charged residues (sticks) facing the interface as marked by the dashed gray rectangle in (**b**).

The octameric ring exhibits pronounced charge anisotropy, with positive (P-side) and negative (N-side) charge predominating on the opposite surfaces (Fig. 2d), which explains the ionic strength-dependent size distribution of the self-assembled particles (Supplementary Fig. 6b).

**Single amino acid substitutions at the N-side restore filament formation.** To investigate how disturbance of electrostatics affects the interactions between the H2T double rings, we substituted individual nonconserved amino acids in the core region pointing to the interface between the two trCP rings (Fig. 2e; Supplementary Fig. 7).

The substitutions on the N-side of the ring, trCP$^{L99C}$, trCP$^{K153E}$ and trCP$^{E150C}$ showed a markedly increased tendency to form RNA-free assemblies larger than double rings (Fig. 3a–d; Supplementary Figs. 8, 9). The formation of double rings was negligible in the case of trCP$^{L99C}$ and trCP$^{K153E}$, and instead we observed the formation of exclusively RNA-free filaments with helical (predominant form) or stacked ring architecture (Fig. 3b, c; Supplementary Fig. 8). In the case of trCP$^{E150C}$, the assortment of particles was more heterogeneous, ranging from double rings to filaments. trCP$^{E150C}$ filaments accounted for only around 45% of the observed particles, with helical and stacked ring architectures represented to a similar extent (Supplementary Fig. 9). Interestingly, among various types of particles in the rCP$^{E150C}$ sample, we detected a significant proportion of two novel architectures (Fig. 3d). One of them with a central cube-shaped body composed of six orthogonally arranged octameric rings growing outwards by stacking copies of the rings to form cross-shaped junctions (Fig. 3d middle). In the second architectural type, the octameric rings joined to form a central spherical body on whose surface additional rings stacked in at least one direction (Fig. 3d right; Supplementary Fig. 9c). Overall, single amino acid substitutions of selected nonconserved residues

on the N-side increased the stickiness of the surface and restored the formation of filamentous assemblies.

**Single amino acid substitutions at the P-side lead to the formation of flipped double rings, cubic and spherical particles.** Single amino acid substitutions at the P-side led to the formation of architecturally more homogeneous particles (Fig. 4a; Supplementary Fig. 10). trCP$^{K176E}$, trCP$^{G193D}$ and trCP$^{G193C}$ assembled exclusively into double octameric rings, with cryo-EM reconstruction of trCP$^{K176E}$ revealing a head-to-head (H2H) arrangement of the two rings (Fig. 4b, c; Supplementary Figs. 10a, b, 11). Again, the two rings are held together by His$_6$-tags (Supplementary Fig. 11c–f), but their central axis is slightly tilted compared with the trCP H2T double rings (Fig. 4c). However, no further stacking of H2H double rings or formation of filaments was detected, possibly due to the fact that both N-side surfaces in the double ring are exposed to the exterior.

SEC analysis of the P-side mutants trCP$^{K176C}$, trCP$^{K176S}$, and trCP$^{K177E}$ indicated the formation of larger particles than double rings (Fig. 4a; Supplementary Figs. 12, 13). Cryo-EM 2D class averages showed that most of these particles had a cubic shape (Fig. 4b; Supplementary Fig. 10c). 3D reconstruction of trCP$^{K176C}$ with an overall resolution of 3.0 Å revealed the cubes consisted of six orthogonally arranged rings (Fig. 4c; Supplementary Fig. 12), with no additional stacking of rings as found in trCP$^{E150C}$ (Fig. 3d middle). In the case of trCP$^{K177E}$, around 30% of the particles had a spherical shape, consisting of 9 rings (Fig. 4c; Supplementary Fig. 13), similar to the spherical core of the particles formed by trCP$^{E150C}$, but again without further stacking of rings on the exposed N-side.

Overall, selected amino acid substitutions of the nonconserved residues on the P-side facilitated the association of the octameric rings with the P-side involved in the interactions and N-sides exposed to exterior. This led to the formation of smaller particles,

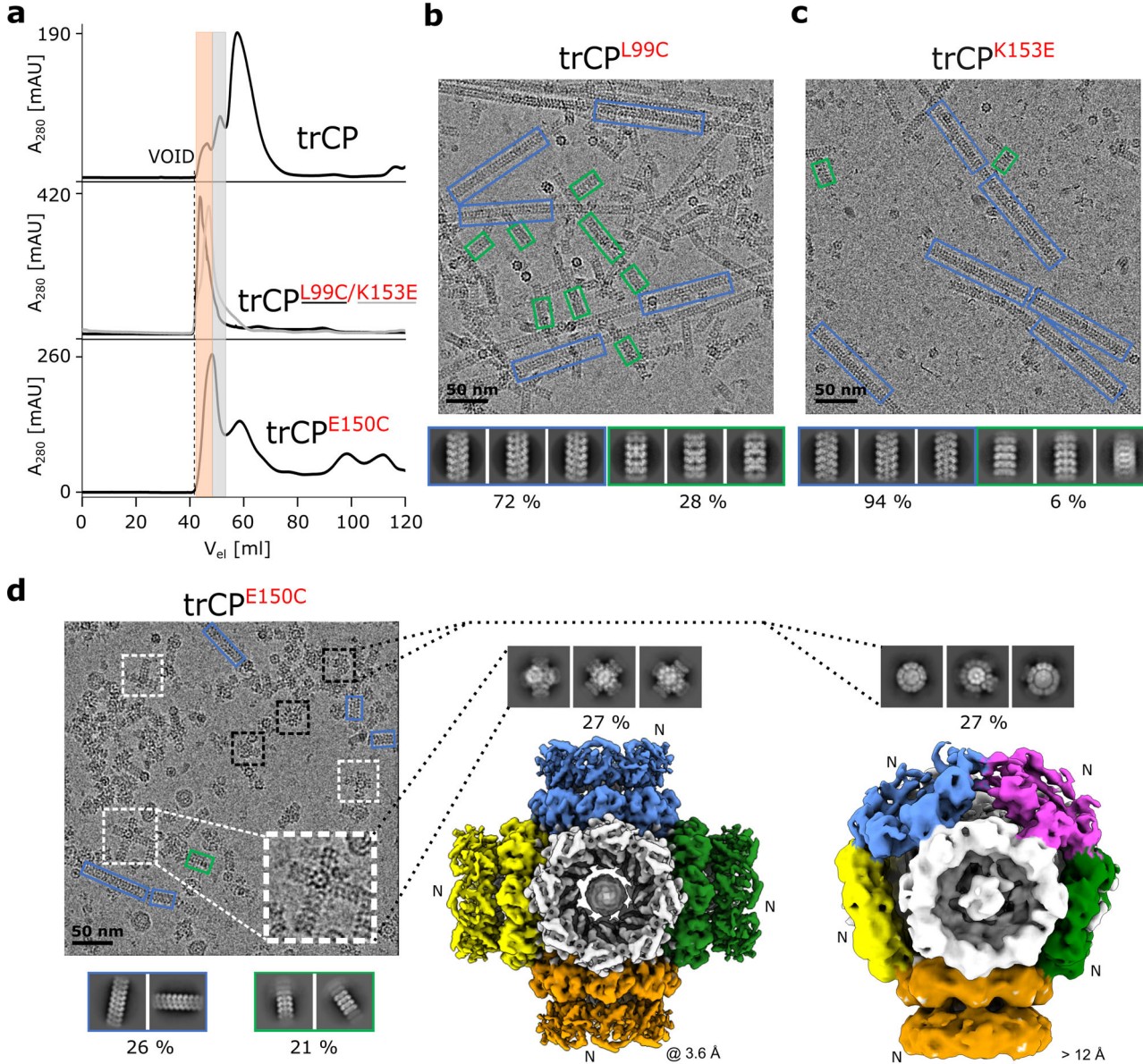

**Fig. 3 N-side mutations resume formation of filaments and lead to novel architectures of filament junctions. a** Comparison of SEC analysis (HiLoad Superdex 200 16/600) between the trCP sample and its N-side mutants trCP[L99C], trCP[K153E], and trCP[E150C]. The orange and gray shading indicate distinct fractions eluting earlier than H2T double rings formed by the trCP, thereby indicating formation of larger particles. The micrographs in **b–d** show samples before SEC analysis. **b, c** Cryo-EM micrograph and 2D class averages of trCP[L99C] (**a**) or trCP[K153E] (**b**) filaments. Blue rectangles: helical RNA-free filaments; green rectangles: RNA-free stacked-ring filaments. Percentage of each type of filamentous particle is indicated. **d** Left: Cryo-EM micrograph of trCP[E150C] sample with 2D class averages of filaments below. White and black dashed rectangles: filament junctions growing form the central cubes or spheres, respectively; blue rectangles: RNA-free helical filaments; green rectangles: RNA-free stacked-ring filaments. Middle and right: 2D class averages and 3D reconstructions of trCP[E150C] filament junctions growing from the central cubic (cross-shaped junctions, middle) or spherical (spherical junctions, right) arrangement of octameric rings, respectively, with corresponding overall resolutions in Å. "N" denotes the N-side of the rings.

such as H2H double rings, and cubic or spherical assemblies of rings.

**The cubic particles are stabilized by hydrophobic interactions and contain CP-derived cargo.** To better understand what drives the orthogonal assembly of the trCP-derivatives, we analyzed the interactions in the locally refined 3D reconstruction of the trCP[K176C] cubes of 3.2 Å resolution (Supplementary Fig. 12; Table 1). This revealed that the hydrophobic interactions between the P-sides of the ring pairs on the C2 symmetry axis are crucial for stable assembly (Fig. 5a; Supplementary Fig. 14a). Two sub-units of each interacting ring contribute to stabilization, one

through the residues of the α5-helix from the core and the other through N-IDR residues, including the α1-helix (Fig. 5a). No disulfide bond was observed between the rings, because the C176 residues in the adjacent rings are too far apart. M54 in N-IDR is crucial for maintaining the interactions, as replacement by Cys in trCP[M54C+K176C] resulted in the formation of H2H double rings instead of cubes (Supplementary Fig. 14b, c).

The center of each octameric ring contained a blob of density, which disappeared after removal of the His$_6$-tags (trCP[K176C-noHis]) without affecting the cubic architecture (Fig. 5b; Supplementary Fig. 15a). Another blob of density was observed in the center of all cubic assemblies (Fig. 5c; Supplementary

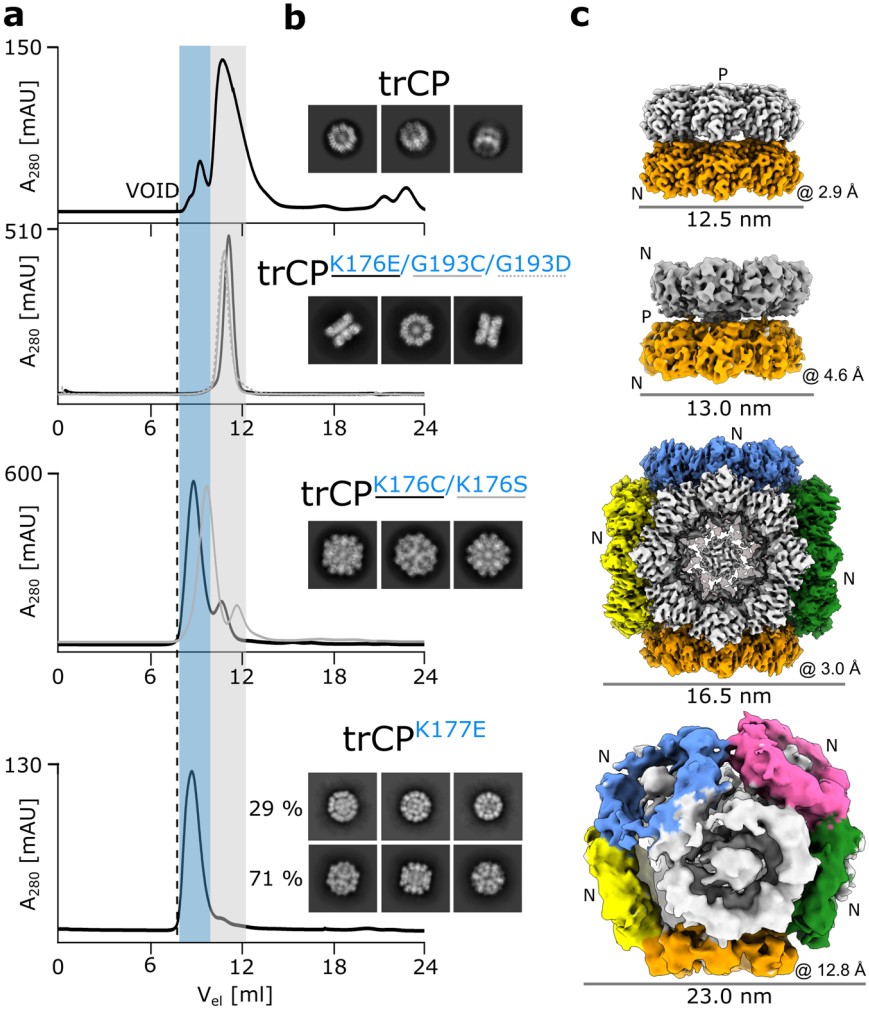

**Fig. 4 P-side mutations lead to novel octameric-ring assemblies, such as H2H double rings, cubes and spheres. a** Comparison of SEC (Superdex200 10/300 GL) analysis of trCP and its P-side mutants trCP$^{K176E}$, trCP$^{G193C}$, trCP$^{G193D}$, trCP$^{K176C}$, trCP$^{K176S}$ and trCP$^{K177E}$. Gray and blue shading indicate SEC fractions that elute similarly to the H2T rings and at an earlier time point, respectively. **b** 2D class averages and **c** 3D reconstructions (colored by rings) are shown from top to bottom for trCP, trCP$^{K176E}$, trCP$^{K176C}$ and trCP$^{K177E}$ with their overall resolutions (Å) and particle diameter (nm) indicated below each reconstruction. "N" and "P" denote the N- and P-sides of the rings. For clarity, only the 3D reconstruction of the spherical trCP$^{K177E}$ assembly is shown, 3D reconstruction for cubes can be found on Supplementary Fig. 13.

Fig. 15b), indicating the presence of putative cargo. Native mass spectrometry analysis of trCP$^{K176C}$ (Supplementary Fig. 15c) revealed charge series around 10,000 m/z, consistent with a double ring, and unresolvable peaks around 18,500 m/z, that we assign tentatively to a fully-formed cubic particle. To circumvent the challenge posed by this heterogeneity, we obtained mass photometry data for trCP$^{K176C}$, trCP$^{K176C-noHis}$ and trCP$^{K176S}$ (Supplementary Fig. 15d–f). We measured masses of ~1.3 MDa for each, a mass higher than expected based on 48 copies of the protomers and consistent with a central cargo of approximately 250–350 kDa (Fig. 5c; Supplementary Fig. 15d, f). When the cubes were disassembled under denaturing conditions, no significant impurities were identified in the denatured spectra beside mass of the monomer (Fig. 5d; Supplementary Fig. 15e, g). A small population of covalently associated dimer was also present only in trCP$^{K176C}$ and trCP$^{K176C-noHis}$. However, these CP dimers did not originate from the octameric rings assembling the cubes, as no disulfide bonds were observed within or between the octameric rings (Fig. 5a). Although we cannot assess at this point how important the central protein mass is for self-assembly, our results clearly indicate that no molecular species other than

the subunits of trCP mutant are required for the formation of these cubic particles.

Due to highly symmetrical distribution and exposure of the C-termini on the surface of the octameric rings in the cubes, we replaced the C-terminal His$_6$-tag on trCP$^{K176C}$ with the SpyTag[30] (trCP$^{K176C-SpyTag}$). Cubic particles, similar to those with C-terminal His$_6$-tags were formed (Supplementary Fig. 16).

Given the rather unexpected result of self-assembly of the trCP-mutants into cubes, we investigated whether the preferential orthogonal assembly of the K176C mutant compared with the wild type trCP could be predicted by the coarse-grained molecular dynamics simulations (Fig. 5e). Starting from randomly distributed octameric trCP$^{noHis}$ or trCP$^{K176C-noHis}$ rings in aqueous solution, the trCP$^{K176C-noHis}$ rings were indeed more prone to form orthogonal ring assemblies (triplets) (Fig. 5e; Supplementary Fig. 17) than the nonmutant trCP$^{noHis}$.

In summary, the cubic assemblies of selected P-side trCP mutants are composed exclusively of CP-derived units. 48 surface exposed C-termini can be modified to carry (removable) affinity tags such as His$_6$-tag or Spy-tag.

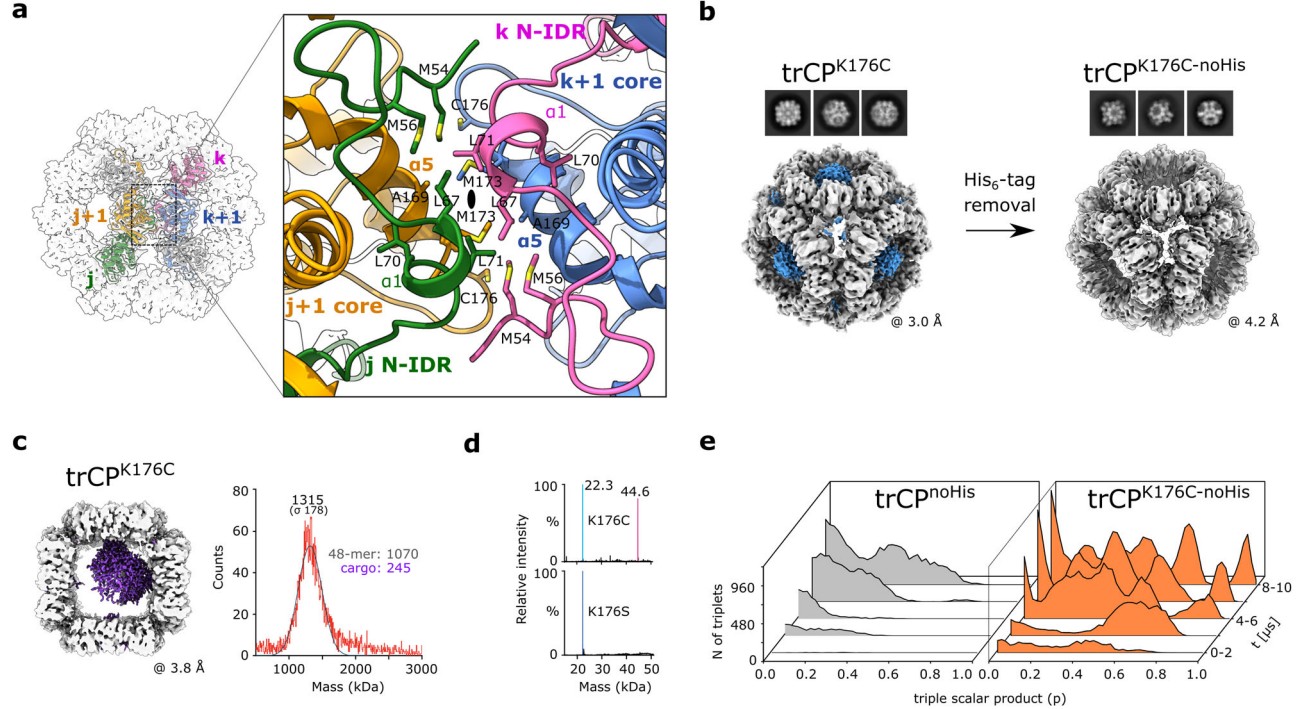

**Fig. 5 The orthogonal assembly of octameric rings into cubes is driven by electrostatics and stabilized by hydrophobic interactions. a** Left: cubic assembly of trCP$^{K176C}$ along the C2 symmetry axis (black oval). Four distinctly colored trCP$^{K176C}$ subunits from adjacent rings are shown in colored ribbons (j, j+1 in one ring; k, k+1 in the adjacent ring). Gray surface: cryo-EM density. Right: magnification of the contact between two rings formed. Hydrophobic residues are shown in sticks. **b** Cryo-EM 2D class averages and 3D reconstructions of trCP$^{K176C}$ cubes before (left) and after (right) His$_6$-tag removal, with the density map corresponding to His$_6$-tag clustering in trCP$^{K176C}$ cubes shown in blue. **c** Left: asymmetric cryo-EM reconstruction of trCP$^{K176C}$ cube with additional central density in purple. The density maps of the front and back rings have been removed for clarity. Right: mass photometry spectra of the entire trCP$^{K176C}$ assembly show a peak centered around 1.3 MDa – consistent with additional cargo of about 245 kDa. **d** Deconvolution of denaturing mass spectrometry data for trCP$^{K176C}$ and trCP$^{K176S}$ reveals monomer masses of 22.3 kDa. Mutation of cysteine to serine eliminates the population of dimers. **e** Time evolution of clustering of triples of interacting rings for trCP$^{K176C-noHis}$ and trCP$^{noHis}$ during MD simulation as a function of shape denoted by a triple scalar product p of ring orientations. Orthogonal packing corresponds to p = 1, while planar packing corresponds to p = 0. The source data are provided in the Supplementary Data file.

**Self-assembly can be controlled by fusion of heterologous proteins with CP.** The supramolecular assemblies described above were purified directly from bacterial cell lysates. Next, we developed a system to prevent the self-assembly process in the expression system and instead trigger it in a controlled environment in vitro. To prevent the formation of filaments with C-IDRs packed in the lumen of the filament (Fig. 1c), we fused the 43-kDa maltose-binding protein (MBP) to the C-terminus of CP (Fig. 6a). Indeed, the CP-MBP fusion did not form filaments, and we were able to isolate the monomeric CP-MBP units (Supplementary Fig. 18a). The purified monomeric fraction of CP-MBP was then exposed in vitro to the tobacco etch virus (TEV) protease, which released MBP from CP, resulting in the formation of RNA-free filaments (ivVLP$^{WT}$) (Supplementary Fig. 18b, c). We then applied this procedure to the CP$^{\Delta C40}$ fusion with MBP (CP$^{\Delta C40}$-MBP). In contrast to the VLP$^{\Delta C40}$ formed in bacteria (Fig. 1e), the filaments produced in vitro were architecturally nearly homogenous, with 97% of the RNA-free stacked-ring architecture (ivVLP$^{\Delta C40:r}$) (Fig. 6b; Supplementary Fig. 18d, e). Furthermore, this concept was successfully used for the in vitro triggered assembly of nanocubes, ivtrCP$^{K176C}$ (Fig. 6c; Supplementary Fig. 18f–h). Also in this case, cryo-EM 3D reconstruction revealed a cargo in the center of the cubes (Supplementary Fig. 18h).

In summary, spontaneous self-assembly of CP and its derivatives in the bacterial expression system can be prevented by fusion of a heterologous protein at their C-termini. In vitro

triggered self-assembly by proteolytic release of the fused protein leads to the formation of highly ordered RNA-free nanoparticles.

**VLPs can be further stabilized by introducing disulfide bonds between CPs.** It has already been shown for filamentous protein or peptide self-assemblies[31–33] that the introduction of Cys residues at the interfaces axially connecting the subunits increases the stability of such particles. To investigate this possibility in the case of flexible PVY VLP filaments, we introduced disulfide bonds between adjacent CP subunits based on the VLP$^r$ structural model (Fig. 7a). The double Cys mutants of the full-length CP, T43C+D136C, L99C+K176C, E150C+G193C and S39C+E72C, successfully formed VLPs (Supplementary Fig. 19a) with SDS-PAGE analysis indicating disulfide bond formation (Fig. 7b). With the exception of VLP$^{T43C+D136C}$, these filaments had longer median lengths (Fig. 7c), and elevated melting temperatures for 5–10 °C (Supplementary Fig. 19b, c) compared with the wild type VLPs. Moreover, VLP$^{L99C+K176C}$, VLP$^{E150C+G193C}$, and VLP$^{S39C+E72C}$ filaments survived incubation at 60 °C for 10 min under oxidizing conditions but not under reducing conditions, whereas VLP and VLP$^{T43C+D136C}$ disintegrated in both cases (Fig. 7d). VLP$^{L99C+K176C}$, VLP$^{E150C+G193C}$, and VLP$^{S39C+E72C}$ filaments were structurally polymorphic (Supplementary Fig. 20), and exhibited similar architecture to wild type VLPs, except that the VLP$^{h+RNA}$ form was essentially negligible. We could confirm the formation of disulfide bonds between adjacent rings only in

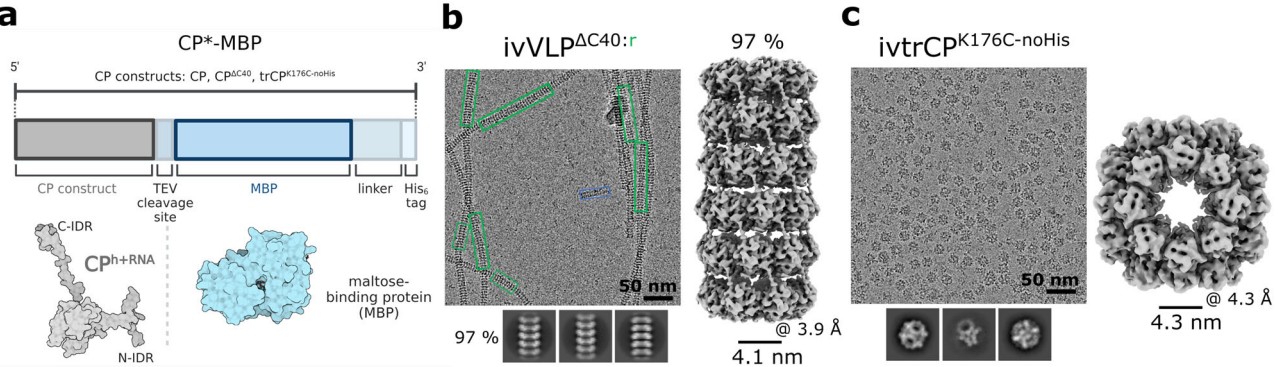

**Fig. 6 In vitro triggered self-assembly of engineered nanoparticles. a** Schematic representation of fusion protein with MBP (CP\*-MBP) with surface representation of both fusion components below (MBP PDB ID: 3HPI). '\*' marks different CP constructs. Created with BioRender (Biorender.com). Micrographs, 2D class averages and 3D reconstructions of in vitro assembled hollow nanotubes with predominant stacked-ring architecture in ivVLP$^{\Delta 40:r}$ (**b**) or cubic particles of ivtrCP$^{K176C\text{-noHis}}$ (**c**). The overall resolution of the 3D reconstructions (Å) and inner particle diameters (nm) are given below. The blue rectangle in **b** marks the presence of small population (~3%) of particles, whose helical parameters resemble the RNA-free helical form.

the asymmetric reconstruction of their stacked ring forms (Fig. 7e). This suggests that not all adjacent Cys are paired, revealing the quasi-equivalence of subunits in the flexible filaments[34]. However, the uneven distribution of disulfide bonds along the filament could also be, at least to some extent, the result of the extreme sensitivity of disulfide-bonds to electron damage radiation[35]. Nevertheless, such interlocking brought adjacent rings in VLP$^{L99C+K176C:r}$ and VLP$^{E150C+G193C:r}$ 3.2 Å and 2.0 Å closer, respectively, than in VLP$^r$ (Fig. 7f). This was not observed in VLP$^{S39C+E72C:r}$ due to stapling of the CPs by structurally plastic N-IDRs (Fig. 7a). No interconnecting cryo-EM density was observed in the VLP$^h$ filaments, likely due to helical averaging along the filament. Overall, VLPs can be further thermally stabilized by introducing disulfide bonds between selected residue positions at axial CP-CP interfaces.

**Monomorphic RNA-encapsidating VLPs can be generated by a single amino acid substitution at the N-IDR/CP-core interface of adjacent CP units**. Unlike other CP double cysteine mutants, VLP$^{T43C+D136C}$ was unique in having more uniform distribution of filament length and instability at 60 °C (Fig. 7c, d). Cryo-EM revealed exclusively RNA-packing filaments with left-handed helical symmetry, and overall resolution of 2.4 Å (Fig. 8a; Supplementary Fig. 21; Table 1), which to our knowledge is the highest resolution for the potyviral VLPs. The structure of CP$^{T43C+D136C}$ and the thermal stability profile of the respective filaments strongly resembled that of PVY virus (Supplementary Fig. 22a, b; Supplementary Table 1).

The cryo-EM density for VLP$^{T43C+D136C}$ was defined starting at residue V44 (Table 1), indicating the absence of the disulfide bond between C43 and C136 and thus the redundancy of one of the introduced cysteines. Indeed, negative staining TEM (nsTEM) of cell lysates revealed that VLP$^{D136C}$ resembled VLP$^{T43C+D136C}$, whereas the purified VLP$^{T43C}$ showed stacked-ring filaments with a length similar to that of wild type VLPs (Supplementary Fig. 22c). Further cryo-EM analysis of purified VLP$^{D136C}$ confirmed monomorphic RNA-packing filaments (Supplementary Fig. 22d).

D136 is located in the β-hairpin of the CP core region. Together with E139 from the same β-hairpin and R46 from the N-IDR of the adjacent CP, it forms a triangle of conserved charged residues (Fig. 8b; Supplementary Fig. 7). Replacement of either residue by Ala resulted in the exclusive (VLP$^{R46A}$) or predominant (VLP$^{E139A}$) formation of RNA-encapsidating filaments (Fig. 8b; Supplementary Fig. 22e, f). In RNA-free

VLPs composed of wild type CP, each CP subunit is linked to four adjacent subunits, with N-IDRs acting as clutches (Supplementary Fig. 2c). Disruption of these interactions by mutations in the R46/D136/E139 triangle favors the VLP$^{h+RNA}$ type of assembly, in which the loss of interaction between the β-hairpin and the N-IDR is compensated for by the extensive interaction network between 13 CP subunits and CP-RNA interactions present in 'h+RNA' filaments. Therefore, it was not surprising that the CP-construct CP$^{\Delta C60:T43C+D136C}$, which integrates both the inability to bind RNA and the weakened N-IDR binding, was not soluble (Supplementary Fig. 22g). These results demonstrate that we can produce monomorphic RNA-encapsidating VLPs with a narrow length distribution by simple modifications of the CP-CP interface at the N-IDR-core contact.

**CP encapsidates ssRNA with limited specificity**. Within the narrow length distribution range of VLP$^{T43C+D136C}$ filaments, we detected four distinct maxima. The first was at 61 nm, which is close to the theoretical length of filaments (65 nm) encapsidating the 807 nt long CP$^{T43C+D136C}$ coding sequence (CDS) (Fig. 8c). Others were at 134 nm, 199 nm, and 267 nm approximately multiples of the first. This could be due to longitudinal fusion of the filaments, as is commonly in potyviruses such as PVY (Supplementary Fig. 23) or potato virus A[36]. Previous studies already suggested that recombinant potyviral CP encapsidates its own mRNA[27,37]. To verify this, we performed analysis of RNA extracted from VLP$^{T43C+D136C}$ filaments. Using RNA-free VLP$^{\Delta C60}$ as a control we showed that 98% of RNA recovered from the purified VLP$^{T43C+D136C}$ sample was the RNA extracted from the filaments (Supplementary Fig. 24a). Reverse transcription quantitative PCR (RT-qPCR) (Supplementary Fig. 24b) showed that CP$^{T43C+D136C}$ mRNA was present at much higher levels in comparison to idnT background gene, reported to be stably expressed in E. coli upon heterologous protein overexpression[38]. To obtain a quantitative overview of all RNA transcripts encapsidated in VLPs, we employed nanopore direct RNA sequencing. This showed that ~70% of the RNA packaged in VLP$^{T43C+D136C}$ belonged to CP$^{T43C+D136C}$ mRNA and 30% were assigned to the bacterial RNAs (Fig. 8d; Supplementary Fig. 24c). Among all coding sequences (CDS), CP$^{T43C+D136C}$ was strongly predominant (~75%), with roughly even coverage of the entire sequence (Supplementary Fig. 24d, e). Some bacterial genes, such as hns, were also detected to a significant extent (11.6%) (Fig. 8e; Supplementary Fig. 24d).

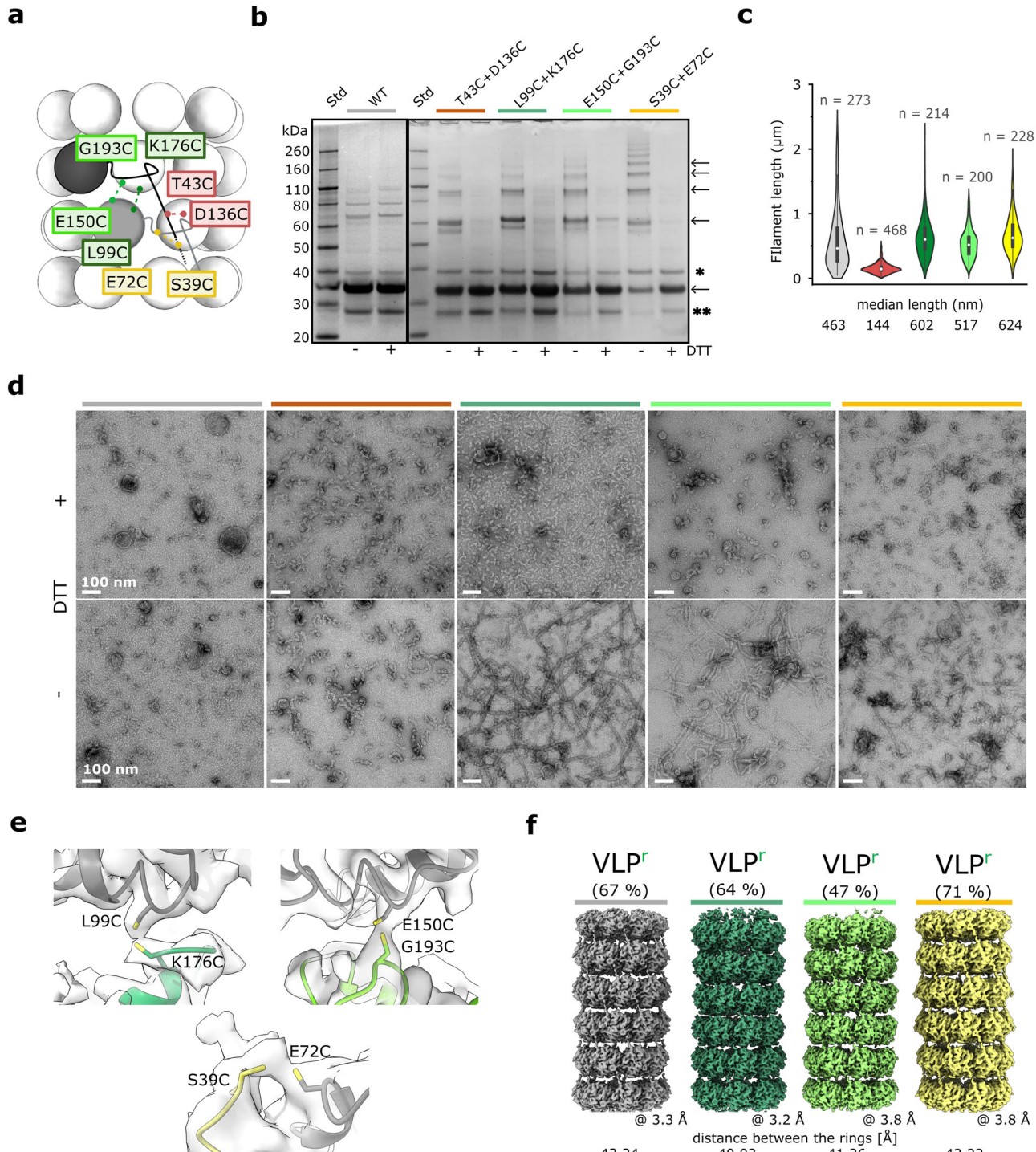

**Fig. 7 Stabilization of VLPs by introducing disulfide bonds between CP units in filaments. a** Schematic representation of VLPs (stacked-ring architecture). Positions of amino acid residues mutated to Cys are indicated in colored rectangles. Pairs of residues simultaneously mutated to Cys and possibly forming disulfide bonds are color-coded. **b** SDS PAGE gels of double Cys mutants (colored as in **a**) under reducing (+DTT) or oxidizing (−DTT) conditions. The bands corresponding to the mutant CP (monomer, oligomers) are indicated with arrows. '*' and '**' indicate impurities. **c** Violin plot showing the length distribution of the filaments, with the corresponding median lengths shown below. 'n': the number of measured filaments. Wild type VLP (gray), VLP$^{T43C+D136C}$ (red), VLP$^{L99C+K176C}$ (dark green), VLP$^{E150C+G193C}$ (light green), VLP$^{S39C+E72C}$ (yellow). **d** Negative staining TEM (nsTEM) micrographs of wild type and double Cys-mutated VLPs (color codes as in **a**) after 10' incubation at 60 °C under oxidizing (−DTT) and reducing (+DTT) conditions. The scale in the nsTEM micrographs represents a spacing of 100 nm. **e** Cryo-EM density at positions expected for disulfide bonds, observed in asymmetric cryo-EM density maps of VLP$^{L99C+K176C}$, VLP$^{E150C+G193C}$, and VLP$^{S39C+E72C}$ stacked ring filaments (color codes as in **a**) with a corresponding mutant model of CP$^{r}$ fitted into the density. **f** Comparison of cryo-EM reconstructions of stacked-ring filaments (VLP$^{r}$) of wild type VLP (gray), VLP$^{L99C+K176C}$ (dark green), VLP$^{E150C+G193C}$ (light green), and VLP$^{S39C+E72C}$ (yellow). The overall resolution and the distances between adjacent rings are shown. The source data for panels b and c are provided as Supplementary Data file.

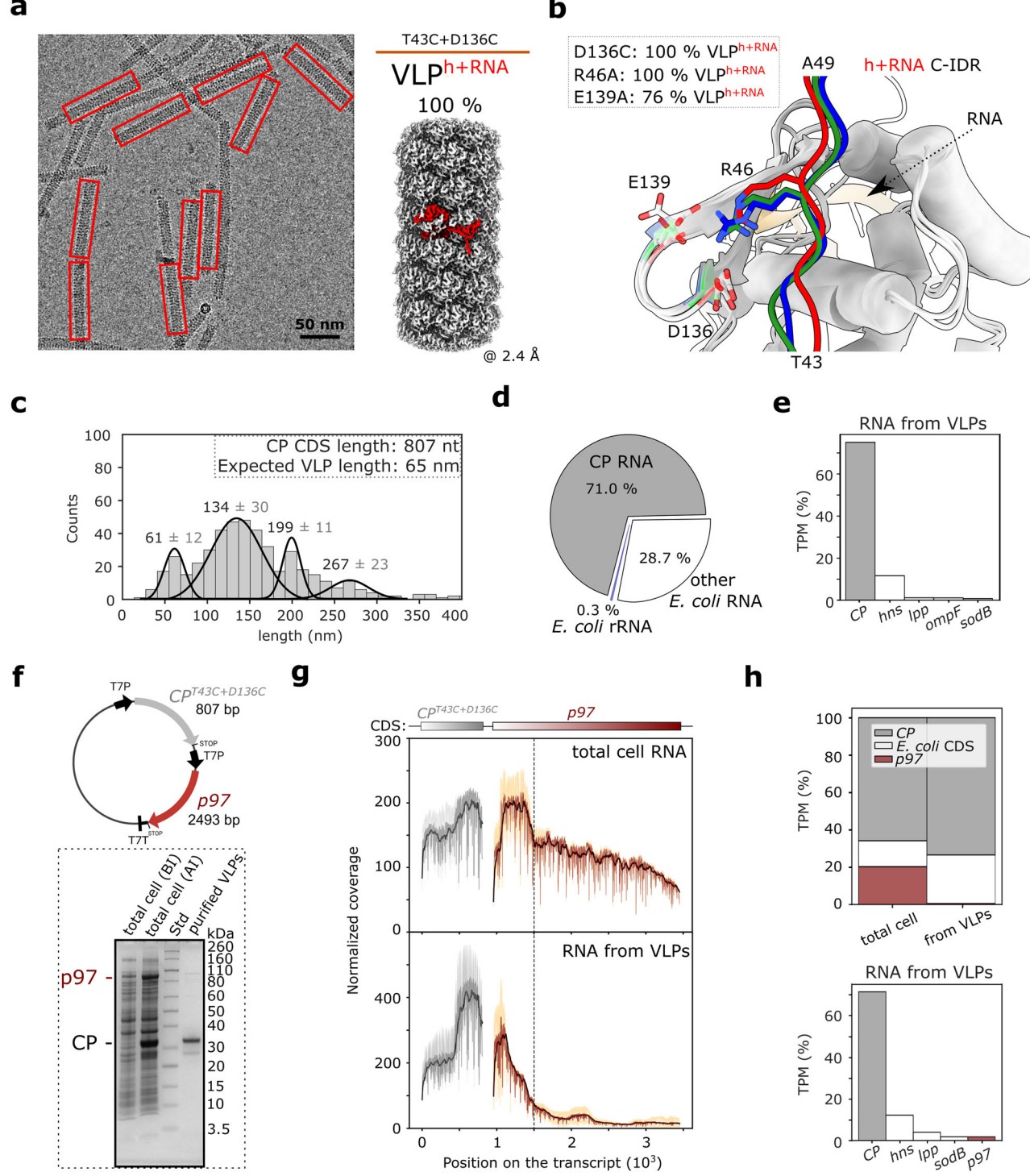

These experiments suggest that the specificity of RNA encapsidation by CP is limited. Next, we investigated, whether we could encapsidate the mRNA of interest into the filaments formed by CP$^{T43C+D136C}$. As a heterologous gene of interest, we chose the gene encoding p97, a human protein forming ~600 kDa hexamers[39] that differ in architecture from VLPs. We first attempted to encapsidate *p97* mRNA by adding in vitro transcribed mRNA to the CP-MBP system described above. However, after the release of MBP, no RNA was encapsidated, likely due to secondary or tertiary structural elements in the RNA produced in vitro that prevent CP from self-assembling around it. To address this issue, we used a bacterial co-expression system so

that the nascent *CP* mRNA and the *p97* mRNA were produced in temporal and spatial proximity (Fig. 8f). In this system, two heterologous mRNAs are transcribed, one for CP$^{T43C+D136C}$ and p97 (*CP+p97*) and the other for p97 only (*p97*). CP$^{T43C+D136C}$ and p97 proteins were successfully produced (Fig. 8f), and the VLP purification protocol allowed successful separation of filaments from p97 hexamers (Supplementary Fig. 25a, b).

RNA transcripts were identified and quantified by nanopore direct RNA sequencing of RNA extracted either from cells (total cell RNA) or from purified VLPs (RNA from VLPs). This initially showed enrichment of *CP*, *p97* and some bacterial CDS in VLPs compared with total cell RNA (Supplementary Fig. 25c).

**Fig. 8 Analysis of ssRNA packaged in VLPs formed by CP$^{T43C+D136C}$. a** Left: cryo-EM micrograph of VLP$^{T43C+D136C}$. Only VLPs encapsidating ssRNA were detected (red rectangles). Right: 3D reconstruction of VLP$^{T43C+D136C}$ showing a CP subunit in red. **b** Superposition of CP$_n$ core regions of VLP$^r$, VLP$^h$, and VLP$^{h+RNA}$ with N-IDRs from CP$_{m-2}$ (VLP$^r$, green), CP$_{n-9}$ (VLP$^h$, blue) and CP$_{n-10}$ (VLP$^{h+RNA}$, red) (Supplementary Fig. 2c). RNA in CP$^{h+RNA}$ is shown as an orange cartoon. The conserved residues R46 and D136/E139 are shown in opaque or transparent sticks, respectively, using the same color code as for the N-IDRs. **c** VLP$^{T43C+D136C}$ length distribution of 468 selected particles from the nsTEM micrographs, with values above the peaks indicating the mean length ± SD. Expected VLP length was calculated with helical parameters for VLP$^{T43C+D136C:h+RNA}$ with 5 nt per CP unit. **d** Pie chart showing the mean percentage of RNA sequencing reads (per base) mapped either to *CP* mRNA, *E. coli* rRNA or other *E. coli* RNAs for the RNA extracted from VLP$^{T43C+D136C}$. **e** Histogram showing five most abundant CDS after mapping to 3' ends (Methods) in the RNA extracted from VLP$^{T43C+D136C}$. Shown is the mean value of transcripts per million (TPM, expressed in percent) from two biological replicates. **f** Top: schematic of pRSFDuet-1 vector with introduced *CP$^{T43C+D136C}$* and *p97*. "T7P" and "T7T" designate T7 promotor and terminator, respectively. Bottom: SDS-PAGE of total cell samples before (BI) and after (AI) induction of CP-p97 co-expression, and purified VLPs. **g** Normalized coverage plots of RNA sequencing of *CP* (gray) and *p97* (red) coding sequences of total cell RNA (top) and RNA extracted from VLPs (bottom). Shown are smoothed (black) and raw (dark gray and dark red for CP and p97, respectively) mean of coverage (n = 2); standard deviation is indicated with light gray and orange. The vertical line designates position with significant decrease in normalized coverage in both samples. **h** Top: histogram showing mean values of transcripts per million (TPM, in percent) of mapped reads for *CP* (gray), *p97* (red), or *E. coli* coding sequences (white) for total cell RNA and RNA from VLPs as determined by RNA sequencing. Bottom: histogram showing five most abundant CDS after mapping to 3' ends (Methods) in RNA from VLPs. Each histogram shows the mean values of two biological replicates. The source data for panels c and f are provided as Supplementary Data file.

However, a detailed analysis of sequencing coverage along the *CP* and *p97* CDS revealed important differences between the two samples. Namely, whereas reads mapping to *CP* CDS were very abundant in both samples (Fig. 8g), coverage of *p97* was significantly lower in RNA from VLPs. We also noted a marked decrease in *p97* coverage after position 1500 in the total cell RNA sample (Fig. 8g, vertical dotted line). A sharp decrease was also observed at this position in the RNA from VLPs, with virtually no coverage in the following region, indicating the absence of the full-length *p97* sequence in purified VLPs (Fig. 8g; Supplementary Note 1).

Because of the uneven coverage along both transcripts (Fig. 8g), we performed CDS quantification with the coverage near the 3' end as a sensor for the level of the full-length sequence (Methods). This confirmed a very low abundance of full-length *p97* transcripts in VLPs (1.8%) despite relatively high levels of *p97* mRNA (19.8%) in total cell RNA (Fig. 8h; Supplementary Fig. 25d). Thus, coupling of the synthesis of *p97* mRNA with the production of CP did not result in efficient encapsidation of *p97* mRNA in VLPs in bacteria. Interestingly, bacterial *hns* mRNA was highly represented in VLPs (12.2%), significantly higher compared with its presence in the total cell RNA (3.7%). However, such enrichment in VLPs was not observed for *CP* or *p97* (Supplementary Fig. 25d). Overall, the specificity of the recombinant PVY CP for the encapsidated RNA is not limited to CP mRNA.

## Discussion

Powerful methodological approaches to high-resolution analysis have helped to discover that symmetric supramolecular assemblies of many viruses, storage or transport cages[14], cytoskeleton[40–42], flagella[43,44], amyloid fibers[45] and others can exist in structurally polymorphic states, with each type of self-assembly usually associated with a specific biological function. Structural polymorphism can also be applied to many recombinant VLPs, protein cages, and (artificial) peptide assemblies, providing a large repertoire of molecular platforms for vaccines, drug delivery systems, nanoreactors, biomaterials or nanomachines[14,19,21,46–48]. In particular, CPs from plant viruses represent a great resource for such nanoparticles, as they are biodegradable and usually nonpathogenic to mammals[49]. Among them, the most studied are ssRNA viruses such as rod-shaped TMV, and icosahedral cowpea chlorotic mottle virus (CCMV), whose CPs represent highly tunable molecular platforms for the production of nucleoprotein assemblies with remarkable architectures and material properties[20,21,49].

In this study, we show how the intrinsic structural plasticity of CP from the flexuous filamentous virus PVY enables the formation of a wide assortment of highly-ordered nanoparticles, whose structural and chemical properties can be tailored by simple modifications. Unlike rigid rod-shaped TMV nanoparticles, which generally require an RNA template for stable formation[21], most of the PVY CP types of nanoparticles shown here self-assemble without a template.

Our results can be summarized in seven points. First, recombinant PVY CP can simultaneously form three architecturally distinct types of VLPs (Fig. 1b, c). These filaments are mostly RNA-free, of either stacked-ring or helical architecture, with only a small fraction of the RNA-encapsidating filaments resembling the native virion. We have shown that the major source of structural plasticity and consequently polymorphism is provided by both IDRs and the conserved RNA binding loop S125-G130. The low proportion of RNA-encapsidated VLPs suggests that the efficiency of assembly of RNA-free filaments is higher than that of RNA-encapsidating ones at given conditions. The RNA-free filaments with stacked-ring architecture are predominant and thus represent the most stable form of CP self-assembly. Interestingly, only a slight change in N-IDR conformation leads to the formation of another type of RNA-free filaments with left-helical symmetry. The three types of polymorphic VLPs formed by the wild type PVY CP could potentially mimic different CP assemblies of structurally liable helical virions during different phases of the viral life cycle, such as virion assembly or disassembly and viral cell-to-cell or long-distance transport[11,50], however, future in-depth studies of virus-associated structures *in planta* are required to confirm this. Second, we showed that most of the RNA packaged in recombinant VLPs was CP mRNA (Fig. 8c–e), which may be due to large amount of CP mRNA due to overexpression in bacteria. However, notable amounts of packaged RNAs in VLPs were of a bacterial origin, with some of CDSs even more enriched in VLPs than *CP* or the eukaryotic gene *p97*, compared with their levels in total cell RNA (Supplementary Fig. 25d). Capability of the potyviral CP to encapsidate heterologous viral RNA under certain conditions in vivo was reported before[51,52]. While recombinant CP shows limited specificity, it is expected that in plants, in order to prevent wasting viral resources, the interplay between the viral and/or host factors is dictating packaging of the viral ssRNA into stable virions[11,53]. More detailed studies are needed to understand whether the limited specificity of recombinant CP is due to the specific nucleotide sequence, RNA length, proximity of freshly overexpressed CP to heterologous RNA molecules, or the combination between these factors.

Third, we show that RNA can be encapsidated in VLPs even in the absence of C-IDR. Potyviral C-IDR has been shown to be critical for viral replication and regulated shift from translation to replication[6,54,55]. Here we show that C-IDR does not play an essential structural role in the filament formation or RNA-encapsidation, however, it does affect the fine structural details in filament architecture (Fig. 1e–i). Fourth, in addition to the ability of wild type PVY CP to simultaneously form filaments of different architectures, this protein and thus its self-assembly, is also highly tunable. Simple modifications in CP lead to a lower degree of polymorphism and even to the formation of monomorphic filaments, or filaments with novel architectures (Figs. 1e–i, 7, and 8a, b). Structure-based design can be used to produce purely RNA-encapsidating VLPs with relatively narrow length-distribution or exclusively RNA-free filaments with broad length-distributions (Fig. 7c). In both cases, the lumen of the filament can either be filled with C-IDRs or hollow in their absence. Fifth, we show that we can achieve a striking change in quaternary structure, i.e. structural metamorphosis, by simple genetic modifications of CP. By deletions and/or single-site mutations, we can reduce or even prevent the filament formation and instead produce single or double octameric rings of CP as well as highly ordered cubic or spherical self-assemblies of these rings, which can be further modified to form into cross shaped forms (Figs. 3–5). Sixth, we show that the outer surfaces of CP-derived nanoparticles, especially double rings, cubes, or spheres, can be equipped with surface exposed affinity tags such as His[6]-tag[56] or Spy-tag[30], thereby providing symmetric platforms for further functionalization (Figs. 2c and 5b, Supplementary Fig. 16). Finally, we have developed a system in which CP-derivatives are fused with a heterologous protein attached to its C-terminus to obtain nanoparticles of enhanced purity, which are assembled under defined and controlled in vitro conditions (Fig. 6). Such fusion proteins with IDRs not engaged in self-assemblies could be used to study molecular interactions between individual CPs and other viral or plant host molecules, such as HCPro[57], Argonaut[58], or RNA[59].

In summary, the intrinsic structural plasticity of PVY CP allows a remarkable structural diversity of its supramolecular assemblies. The high-resolution data obtained in this study and the possibility of structure-based design of nanoparticles with novel architectures and tailored properties make PVY CP an excellent candidate for nanobiotechnological applications, such as vaccine and biosensor development, cargo storage and delivery, medical imaging, or energy and nanostructured materials[49,60]. Bacteria represent a preferred expression system as they allow efficient and cost-effective production of nanoparticles. Although the detailed information on the structural diversity of PVY CP shown here is based on nanoparticles produced in bacteria, it may facilitate future studies on the role of PVY CP in its natural environment.

## Methods

**Molecular cloning of CP variants**. The wild type PVY CP from a complementary DNA (cDNA) of PVY-NTN strain (GenBank accession no. KM396648), and its double deletion mutant without (CP$^{\Delta N49C40}$) or with C-terminal His$_6$-tag, preceded by the TEV protease cleavage site (termed truncated CP, trCP), were previously cloned in vectors pT7-7 (CP) and pET28a (CP$^{\Delta N49C40}$, trCP), respectively[6]. The C-IDR deletion constructs were cloned using classical restriction enzyme-based approach and inserted in pET28a vector. To obtain constructs with introduced mutations, site-directed mutagenesis was performed using inverse PCR method[61,62] with one or two oligonucleotides (nucleotide sequence available upon request).

For CP-MBP constructs, sequence encoding maltose-binding protein (MBP) with a C-terminal N-rich linker and "factor Xa" cleavage site, was obtained from the pMAL-c2X vector backbone. This sequence was inserted between TEV protease cleavage site and His$_6$-tag at the C-terminus of His$_6$-tagged CP construct, cloned previously[6]. Cloning of CP$^{\Delta C40}$-MBP and trCP$^{K176C}$-MBP constructs was done via the Gibson cloning method[63,64] (NEB).

For co-expression experiment, $CP^{T43C+D136C}$ and human $p97$ (kindly provided by Dr. Marta Popović, Ruđer Bošković Institute, Croatia) were cloned in the pRSFDuet-1 dual expression system vector (pRSFDuet1-Cdc45) using PCR and Gibson assembly[63,64]. RNA-packing $CP^{T43C+D136C}$ was cloned into the first multiple cloning site (MCS) and $p97$ into the second MCS, while the connecting region was identical to the commercial pRSFDuet-1 backbone. All sequences were verified by nucleotide sequencing (Eurofins Genomics or GENEWIZ).

**Expression and purification of CP variants**. *E. coli* BL21(DE3) cells, transformed with plasmids containing CP constructs, were grown to an OD$_{600}$ of 0.8–1.2 in 2× YT medium (16 g l$^{-1}$ tryptone, 10 g l$^{-1}$ yeast extract, 5 g l$^{-1}$ NaCl) supplemented with 5 mM MgCl$_2$ and 2 mM CaCl$_2$. Gene expression was induced with 0.1 mM Isopropyl β-D-1-thiogalactopyranoside (IPTG) and the cells were grown overnight at 20 °C.

Non His$_6$-tag variants forming VLPs were purified as described previously[6] with minor modifications and all purification steps done at 4 °C. In brief, the harvested cells were lysed by sonication on ice in phosphate-buffered saline (PBS) (1.8 mM KH$_2$PO$_4$, 10.1 mM Na$_2$HPO$_4$, 140 mM NaCl, 2.7 mM KCl, pH 7.4) and centrifuged at 20,000 × g for 40 min. The lysate was incubated for 30 min in the mixture of 4% PEG 8000 and 500 mM NaCl. Following centrifugation for 30 min at 14,000 × g, the pellet with VLPs was resuspended in PBS by gentle overnight shaking. Remaining solid material was removed by 30 min centrifugation at 35,000 × g. The soluble fraction with enriched VLPs was loaded on 20–60% sucrose density gradient and ultracentrifuged at 117,000 × g for 6 h in a Beckman 50 Ti rotor. All fractions of the gradient were collected and analyzed with SDS-PAGE to identify fractions containing CP. Selected fractions were pooled, dialyzed for 24 h against PBS, concentrated using Amicon Ultra centrifugal filters with a 100-kDa molecular weight cut-off to the final concentration of 1–3 mg ml$^{-1}$ and supplied with glycerol up to 5% v/v (final concentration) before storage at −80 °C.

To achieve higher purity of the VLP samples used for RNA extraction, an additional purification step of ammonium sulfate precipitation[65,66] was implemented before the standard VLP purification procedure described above. After cell lysis and centrifugation, ammonium sulfate was added to the soluble fraction to 15% (w/v) concentration. Following stirring for 30 min, the precipitated proteins were pelleted with 15 min centrifugation at 13,400 × g. The process was repeated in a stepwise manner with 5% (w/v) increase in ammonium sulfate concentration up to final 30% (w/v). The pellets pulled at different concentrations of ammonium sulfate were resuspended in PBS and subjected to SDS-PAGE analysis. Fractions with enriched either p97 or CP, were dialyzed via PD-10 desalting columns and with dialysis tubing (12–14 kDa cut-off), with CP-enriched fraction further purified as described above for filamentous VLP. All steps of the purification procedure were done at 4 °C. Final samples were concentrated to a concentration of 1–2 mg ml$^{-1}$ and stored at −80 °C.

His$_6$-tagged proteins (trCP, CP-MBP) were isolated from cells by sonication on ice in PBS with 10 mM imidazole, followed by 40-min centrifugation at 50,000 × g and Ni-NTA chromatography. The non-specifically bound proteins were washed from the

column and the His$_6$-tagged proteins eluted with PBS containing 300 mM imidazole. The eluted fractions were dialyzed against PBS overnight and concentrated using Amicon Ultra (30-kDa or 100-kDa cut-off), and loaded on the size exclusion column Superdex 200 10/300 GL (24 ml) or Superdex 200 16/60 PG (120 ml) (GE Healthcare) with PBS as the running buffer. Fractions with desired trCP variant on SDS-PAGE, were pooled and concentrated using Amicon Ultra centrifugal filters or Pierce™ Protein Concentrators, both with 100-kDa molecular weight cut-off, to the concentration of 1–7 mg ml$^{-1}$ for various assemblies. For CP-MBP, size exclusion chromatography (SEC) step was performed on HiLoad Superdex 200 16/600 at identical conditions as those used for separation of column manufacturers size standards (GE Healthcare). In the case of trCP, trCP$^{K176E}$ and trCP$^{K176C}$, His$_6$-tags were removed using the TEV protease in 1:10-1:20 (TEV:trCP) molar ratio overnight at 20 °C, followed by the second Ni-NTA chromatography or SEC at room temperature. Fractions containing the proteins with cleaved tags were concentrated to 4 mg ml$^{-1}$ and stored at −80 °C. Sample purity and protein folding were checked with SDS-PAGE and circular dichroism spectroscopy, respectively.

**In vitro self-assembly of VLP filaments and cubic particles**. In all types of CP*-MBP fusions, CP* self-assembly was initiated with the addition of TEV protease to the purified CP*-MBP in a molar ratio of 1:10–1:20 (TEV:CP*-MBP) and left overnight at 4 °C. For CP-MBP, the sample after TEV protease cleavage was loaded onto NiNTA column to separate the cleaved His$_6$-tagged MBP and the non-cleaved CP-MBP fusion from the freshly self-assembled VLPs. For CP$^{\Delta C40}$-MBP, assembled VLPs were purified using the standard VLP isolation procedure described above. For trCP$^{K176C}$-MBP, additional purification was done by SEC using Superdex 200 16/600 (120 ml) column. The purified samples were concentrated with Amicon Ultra centrifugal filters (100-kDa cut-off) with presence of filaments before and after TEV protease cleavage supervised by negative staining transmission electron microscopy (nsTEM) for CP-MBP or cryo-electron microscopy (cryo-EM) for CP$^{\Delta C40}$-MBP and trCP$^{K176C}$-MBP.

**Thermal stability assay**. The thermal stability of the proteins was determined by differential scanning fluorimetry (DSF) at a protein concentration of approximately 0.1 mg ml$^{-1}$ in the presence of 2× SYPRO Orange (Thermo Fisher Scientific)[67]. Samples were subjected to temperatures from 25 °C to 95 °C at a gradient of 1 °C min$^{-1}$. Temperature melting profiles were acquired with LightCycler 480 system (Roche). Samples were measured in triplicates with two independent measurements. Melting temperatures T$_m$ were determined as minimum values from first derivative of the measured data curves in OriginPro2023 (OriginLab). All results are expressed as means ± standard deviation (SD) with their comparison performed by one-way ANOVA (analysis of variance) followed by Tukey's multi comparison test. A value of $p < 0.001$ was considered statistically significant. All source data with detailed statistical analysis are provided in the Supplementary Data file.

**Native-PAGE**. Characterization of the protein assemblies in native conditions was performed on 4–16% Native-PAGE Bis-Tris gels (Thermo Fisher Scientific). Samples were mixed with 4x Native PAGE sample buffer (Thermo Fisher Scientific) and run in the Dark Blue Cathode Buffer (Thermo Fisher Scientific) for 60 min at 150 V and for 40–60 min at 250 V according to the manufacturer instructions. The gels were fixed with 40% methanol (v/v) and 10% acetic acid (v/v) and destained with 8% acetic acid (v/v).

**Negative staining transmission electron microscopy (nsTEM)**. For visualization, the final concentration of CP construct was approximately 1.5–3 μM. Copper mesh grids (SPI Supplies) were Formvar-coated, stabilized with carbon and glow-discharged (EM ACE200, Leica Microsystems). The VLP sample (5–20 μl) was applied to a grid, left to soak for 5 min, blotted, washed and contrasted with 1% (w/v) uranyl acetate (aqueous solution). Grids were imaged at 80 kV by CM 100 transmission electron microscope (Philips), equipped with Orius SC 200 camera (Gatan) and Digital Micrograph software 2.1.1 or by TALOS L120 (Thermo Fisher Scientific), operating at 100 kV, equipped with camera Ceta 16 M and Velox v3.0 (Thermo Fisher Scientific).

**Filament length distribution analysis**. Filament lengths were measured from nsTEM micrographs using the Fiji (ImageJ 1.53c) software suite[68] after manually tracing multiple points along at least 200 flexuous filaments. The violin plots of filament length distribution were produced using OriginPro2023 (OriginLab) with median values and ranges above the 25$^{th}$ and below the 75$^{th}$ percentile designated on the plots with white circle and black rectangular box, respectively. Values 'n' above each violin correspond to the number of measured filaments. Histograms of filament length distribution were analyzed using Gauss distribution fit in Origin2018 (OriginLab) and plotted in MATLAB R2021b (MathWorks) with values above the peaks provided as mean ± SD. All filament length measurements are provided in the Supplementary Data file.

**Extraction of the total cell or VLP-encapsidated RNA**. Extraction of total RNA from cells after induced overnight expression was done using the RNeasy Kit with optimized protocol for extraction from *E. coli* adapted from RNAprotect® Bacteria Reagent Handbook (Qiagen)[69]. Specifically, cell lysis was performed enzymatically using lysozyme from chicken egg white (Sigma) and proteinase K (NEB), followed by standard RNeasy protocol with on-column DNase I treatment (Roche) and final elution in RNase-free water.

RNA extraction from the purified VLP samples was performed based on the previously published protocol of extraction from *Potyvirus* particles[70]. In brief, the sample was incubated in the presence of 1% SDS (w/v) at 55 °C for 5 min, followed by phenol-chloroform extraction. The extracted RNA was then precipitated by the addition of 0.5 initial sample volume of 7.5 M ammonium acetate and 2.5 volumes of cold absolute ethanol at −20 °C for 1 h, followed by 25 min centrifugation at 12,000 × g. After washing the pellet with 70% ethanol (v/v) and air-drying for 15–30 min at room temperature, the precipitated RNA was resuspended in DEPC-treated water and treated with Turbo DNase rigorous protocol (Thermo Fisher) to remove any potential DNA contaminants, followed by isolation with RNA Clean & Concentrator-5 kit (Zymo Research) and storage at −80 °C.

**RNA quantification with reverse transcription quantitative polymerase chain reaction**. RNA was reverse transcribed using random hexamer oligos (IDT) and SuperScript IV reverse transcriptase (Thermo Fisher) following the manufacturer's protocol. After RT, cDNA was diluted 10x and used in the PCR. The final qPCR reaction was performed using Fast SYBR Green master mix (Thermo Fisher) in 6 μM primer mix for either *CP$^{T43C+D136C}$* or *idnT* as a control gene, found to be stably expressed in *E. coli* upon induction of protein overexpression[38]. The reaction was measured using LightCycler 480 system (Roche) with the following conditions: 92 °C for 3 min followed by 40 cycles of 3 s at 92 °C and 30 s at 60 °C. The measurements were

made in 2 biological replicates (for each 3 technical replicates). $C_t$ values were obtained using automatic threshold detection by the software (Roche), mean $C_t$ value and standard deviation were calculated in Origin2018 (OriginLab).

**Polyadenylation, direct RNA sequencing and bioinformatic analysis**. For poly(A)-tailing reaction *E. coli* poly(A) polymerase (NEB) was used. Purified RNA was polyadenylated for 1 min at 37 °C in the following reaction mix: 10 µl total RNA, with 2 µl 10× *E. coli* poly(A) polymerase buffer, 2 µl ATP, 5 µl nuclease-free water and 1 µl *E. coli* poly(A) polymerase (NEB). The reaction was stopped by the addition of 5 µl of 50 mM EDTA. Poly-adenylated RNA was cleaned using 2.5× sample volume of AMPure XP beads (Beckman Coulter) and eluted in 10 µl nuclease-free water.

Direct RNA sequencing was performed using the Direct RNA Sequencing protocol (SQK-RNA002) for MinION adapted to sequencing using a Flongle flow cell (Oxford Nanopore Technologies). For adapter ligation, 1 µl of T4 DNA ligase (Thermo Fisher) was used and SuperScript IV (Invitrogen) was used for reverse transcription. For RNA adapter ligation 4 µl NEBNext Quick Ligation Reaction buffer (NEB), 2 µl RNA Adapter (RMX), 1.5 µl T4 DNA ligase were added and the total reaction volume was brought to 20 µl. Finally, RNA was cleaned using 1× sample volume of AMPure XP beads and eluted in 9 µl Elution Buffer (EB). The eluate was then loaded on a Flongle R9.4.1 flow cell.

Raw read files were base-called using guppy version 6.0.0 using high-accuracy mode (rna_r9.4.1_70bps_hac.cfg) with filtering set to minimum Q-score of 7. Base-called reads were mapped either to *E. coli* genomic coding sequences (CDS) or its genome. In the first case, base-called reads were mapped to *E. coli BL21(DE3)* genomic coding sequences (genome NCBI Reference Sequence NZ_CP081489.1) to which custom coding sequences of *CP* and *p97* from the expression vector pRSF-Duet1 were added manually. Mapping was performed using minimap2[71] with "-ax map-ont -k14" parameters. Mapped reads were filtered using samtools v1.6[72] and mapped reads with the MAPQ score 60 were retained. Reads mapped to *E. coli* CDS were counted using NanoCount[73] with default parameters, where filtering for reads that map within 50 nt of the 3'-end of the reference was enabled (3' filtering). Estimated count values per coding sequence were used to calculate adjusted transcripts per million (TPM) values for only those transcripts that were present in both biological replicates. Values for transcripts not present in both biological replicates were discarded. Mann-Whitney-Wilcoxon two-sided test was performed over adjusted TPM values between both replicates of each measured RNA sample ($CP^{T43C+D136C}$ expression: $p.\ val. = 3.9 \times 10^{-7}$ comparing 'RNA from VLPs' replicates; $CP^{T43C+D136C}$-p97 co-expression: $p.\ val. = 0.027$ comparing 'total cell RNA' replicates, $p.\ val. = 0.204$ comparing 'RNA from VLPs' replicates). Adjusted TPM values were averaged between biological replicates and used further in downstream analyses. Per base coverage was computed using bedtools v2.30.0. Per base coverage was further normalized by division with the sum of coverage of all bases mapped and multiplied by a million bases. Coverage plots were plotted using seaborn Python library. Smoothened per base coverage means were calculated as the mean value of 40 consecutive bases.

Mapping the reads to the *E. coli BL21(DE3)* genome (NCBI Reference NZ_CP081489.1) to which CP and p97 coding sequences were added as additional chromosomes, was performed using minimap2 with "-ax map-ont -k14" parameters. Mapped reads were filtered using samtools v1.6 for the MAPQ score of 60. The amount of rRNA reads in each of the samples was calculated by intersecting the filtered mapped sequences with a genomic GTF file using bedtools intersect. Relative amounts of different RNA species as bp % values were calculated by adding gene-specific base pairs (bps) and comparing them to the sum of all mapped bps.

**Cryo-EM grid preparation and data acquisition**. For grid preparation, 3 µl of the sample with a concentration of around 1 mg ml⁻¹ for the filamentous particles or 3–4 mg ml⁻¹ for non-filamentous assemblies, were applied to glow-discharged Quantifoil 200-mesh R2/2 holey carbon grids (Quantifoil) followed by vitrification in Vitrobot Mark IV (Thermo Fisher Scientific). With the exception of VLP$^{\Delta C40}$ and trCP$^{K176C}$, the samples were imaged on Glacios transmission electron microscope operated at 200 kV and equipped with Falcon 3 direct electron detector (Thermo Fisher Scientific). Data sets were acquired at a nominal magnification of 150,000 corresponding to calibrated pixel size of 0.950 Å and defocus range between −0.8 and −2.1 µm with a total dose of around 40 e⁻ Å⁻².

For VLP$^{\Delta C40}$ and trCP$^{K176C}$, cryo-EM data was collected on Titan Krios transmission electron microscope (Thermo Fisher Scientific) operated at 300 kV at CEITEC, Brno, Czech Republic. VLP$^{\Delta C40}$ data set was acquired in linear mode with Falcon2 (Thermo Fisher Scientific) direct electron detector at a nominal magnification of 75,000 corresponding to a calibrated pixel size of 1.063 Å and defocus range of −1.3 and −0.4 µm, with 40 frames collected within 1.02 s exposure giving a total dose of 84 e⁻/Å². trCP$^{K176C}$ data set was acquired on K2 Summit direct electron detector (Gatan) operating in counting mode at a nominal magnification of 165,000 corresponding to a pixel size of 0.822 Å and defocus range between −0.3 and −3.6 µm. 32-frame movies were collected during 4 s exposure time with a total dose of 32 e⁻ Å⁻².

**Cryo-EM image processing**. The detailed workflow for each dataset-specific reconstruction is presented in Supplementary Figs. 1, 3, 5, 8, 9, 11–13, 20 and 21. In general, more than 500 movies were collected for each sample and used for cryo-EM data processing, performed in cryoSPARC v3.3 or 4.1[74–76] except for VLP$^{\Delta C40}$ and VLP$^{T43C+D136C}$, where RELION-3.1[77] was used.

For disulfide bond-stapled VLPs, cryo-EM reconstructions (Supplementary Fig. 20) were performed using C1 symmetry with additional selection of subclasses based on observed extensive conformational variability using 3D Variability analysis[75] and 3D classification. Final sharpened non-symmetric cryo-EM maps were checked for connecting density.

The resolutions of the final cryo-EM maps, in some cases locally sharpened with DeepEMhancer v0.13[78], were determined based on the gold-standard FSC criterion of 0.143[79]. Local resolutions were calculated using BlockRes[80], and cryo-EM densities were visualized in UCSF Chimera 1.16[81] and ChimeraX 1.5. Details, EMPIAR and EMDB codes are provided in Table 1 and Supplementary Tables 1 and 2.

**Model building**. PDB ID codes of initial models used for model building are provided in Table 1. In each case, the initial model was fitted into the reconstructed cryo-EM map using UCSF Chimera 1.16 with one central CP unit and all the neighbors in direct contact subjected to several iterative cycles of manual refinement using WinCoot 0.9.8.1[82] and real-space refinement with secondary structure and geometry restraints in Phenix 1.20.1 package[83]. For helical filaments with RNA, the segment of 5 uracils from the PVY virion (PDB ID: 6HXX) was fitted into the empty density of one CP and subjected to the same iterative cycles of refinement as for the protein components. Molprobity[84] was used for validation of individual models after each cycle.

For trCP$^{K176C}$, atomic models were built in the cryo-EM map after local refinement (EMD-17063) with two distinct protomer structures, one in C2 symmetric contact and the adjacent one. Model from the locally-refined cryo-EM map (PDB ID 8OPK) was rigid body-docked into the globally-refined cryo-EM map (EMD-17062) to obtain the atomic model of the entire cubic particle (PDB ID 8OPJ). Final 3D models were visualized in UCSF Chimera 1.16 and ChimeraX 1.5[85]. The surface electrostatic potential of the trCP was calculated by APBS 3.4.1[86]. Detailed statistics of model building and refinement are presented in Table 1.

**Mass spectrometry and photometry**. To denature the assembly and determine an accurate monomer mass, constructs trCP$^{K176C}$, trCP$^{K176C-noHis}$ and trCP$^{K176S}$ in PBS were diluted into a solution of 50% acetonitrile and 2% formic acid, to a final concentration of 5 μM of the cubic 48mer. For the native spectrum of trCP$^{K176C}$, the sample had buffer exchanged in 200 mM ammonium acetate (pH 6.9) using Bio-Spin 6 columns (Bio Rad) and sprayed at the same concentration. Nanoelectrospray mass spectrometry data were acquired using a QExactive UHMR mass spectrometer (ThermoFisher) using gold-plated 1.2 OD mm capillaries prepared in-house, as previously described[87]. Resultant spectra were deconvolved and analyzed using UniDec[88].

To acquire mass photometry data[89], the constructs were diluted with PBS to 50 nM and measured on a Refeyn Two$^{MP}$ mass photometer and analyzed using DiscoverMP v2.5.0 (Refeyn Ltd).

**Molecular dynamics simulations (MD)**. We took the atomic model of one ring from CP$^{\Delta C40:r}$ and truncated it on the N-terminus (ΔN49) to simulate the trCP$^{noHis}$. For trCP$^{K176C-noHis}$ starting model, an additional K176C mutation was introduced. We constructed two coarse-grained (CG) systems by arranging eight randomly oriented rings (either trCP$^{noHis}$ or trCP$^{K176C-noHis}$) with a minimum initial pairwise spacing of 16 nm between rings (i.e., 1.5 times the ring diameter). Rings were immersed in a $50 \times 50 \times 50$ nm$^3$ cubic box of water in which neutralizing counterions were eventually added. Each CG model of the ring was generated from the corresponding atomic model by using Martinize2 protocol[90]. An elastic network[91] was applied to maintain the overall internal structure of an individual ring. All CG-MD simulations were performed with GROMACS 2019.6[92] and Martini 3.0 force field[93]. The systems were energy minimized with the steepest descent algorithm (50.000 steps), followed by a brief NPT (keeping the number, pressure and temperature constant) equilibration cycle to relax the initial configurations (200.000 steps of 5 fs). Afterward, the systems were simulated for 10 μs in an NPT ensemble with periodic boundary conditions (500.000.000 steps of 20 fs). The temperature was maintained at 300 K and pressure at 1 bar by coupling the dynamics using V-rescale thermostat[94] and Berendsen barostat[95]. The cut-off value for the Coulomb and van der Waals interactions was set to 1.1 nm, and a relative dielectric constant was set to 15. The rings freely diffuse in the solution until they hit each other and occasionally form a contact. The formation of ring clusters was analyzed by first identifying all ring-triplets sharing all three pairwise contacts to each other for each given instant of time (trajectory frame). As an order parameter revealing the form of the clusters we introduced the absolute value of the scalar triple product $p = |(n_1, n_2, n_3)| = |n_1(n_2 \times n_3)|$, with vector $n_i$ identifying a directional unit vector along the $i$-th ring normal. The value $p = 0$ corresponds to the plane-distributed ring-triplets while $p = 1$ corresponds to the mutually orthogonal orientation of rings forming the corner of the cube. The coordinates for the initial (equilibrated) structures and the final structures (after 10 micro seconds) can be obtained upon request in GROMACS format.

**Reporting summary**. Further information on research design is available in the Nature Portfolio Reporting Summary linked to this article.

## Data availability

Cryo-EM maps and atomic models have been deposited in the Electron Microscopy Data Bank (EMDB) and wwPDB, respectively, with EMDB/PDB accession codes: EMD-17046/8OPA, EMD-17047/8OPB, EMD-17048/8OPC, EMD-17049/8OPD, EMD-17050/8OPE, EMD-17051/8OPF, EMD-17052/8OPG, EMD-17053/8OPH, EMD-17054, EMD-17055, EMD-17056, EMD-17057, EMD-17058, EMD-17059, EMD-17060, EMD-17061, EMD-17062/8OPJ, EMD-17063/8OPK, EMD-17064, EMD-17065, EMD-17066, EMD-17067, EMD-17068, EMD-17069, EMD-17070, EMD-17071, EMD-17072/8OPL, EMD-17073, EMD-17074 and EMD-17075, with corresponding structures and atomic models provided in Table 1 and Supplementary Tables 1 and 2. Raw cryo-EM datasets have been deposited to the Electron Microscopy Pilot Image Archive (EMPIAR) with accession codes EMPIAR-11545 (EMD-17046/8OPA, EMD-17047/8OPB, EMD-17048/8OPC), EMPIAR-11546 (EMD-17049/8OPD, EMD-17050/8OPE), EMPIAR-11547 (EMD-17052/8OPG), EMPIAR-11548 (EMD-17053/8OPH, EMD-17054, EMD-17055), EMPIAR-11549 (EMD-17062/8OPJ, EMD-17063/8OPK) and EMPIAR-11550 (EMD-17072/8OPL). RNA nanopore sequencing data have been deposited on European nucleotide archive (ENA) with accession code PRJEB61146. All data are available in the main text, figures and supplementary information. Source data are provided with this paper as Supplementary Data file. Additional data related to this paper may be requested from the authors.

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

## Acknowledgements

We thank Jelka Lenarčič and Tanja Peric for technical assistance with cloning, Izidor Friškovec and Marija Srnko for technical support with protein purification and biochemical characterization, and Dr. Matic Kisovec for technical support with cryo-EM data analysis. We thank Dr. Miha Modic and Dr. Oscar G. Wilkins for advice on nanopore RNA sequence analysis and Prof. Dr. Jernej Ule for use of nanopore sequencing in his 'RNA networks' laboratory. We thank Dr. Ana Crnković for advice on the dual-expression system and Dr. Ion Gutierrez Aguirre for kindly providing the PVY virus. CIISB research infrastructure projects LM2018127 and LM2015043, funded by MEYS CR, are gratefully acknowledged for financial support of measurements at CF CryoEM. We thank the National Institute of Chemistry Cryo-EM Facility supported by the Slovenian Research Agency Infrastructure Program IO-0003. We thank the Slovenian Research Agency for funding (grant numbers P1-0391 (M.P., A.K.), J1-4410 (M.P., A.K., L.K.), PhD fellowship (N.K.), P4-0165 (M.T.Ž.), J1-2467 (M.T.Ž.), P1-0010 (F.M.)). We thank the National Institute of Chemistry, Slovenia, for the PhD 'Janko Jamnik' fellowships to L.K. and T.K. We thank European Union's Horizon 2020 research and innovation program for the grant 835300-RNPdynamics (T.K., Ž.V.) and the grant from the Biotechnology and Biological Sciences Research Council BBSRC sLoLa BB/W00349X/1 (E.H., J.L.P.B.).

## Author contributions

The study was conceptualized by M.P. and L.K. Molecular cloning was conducted by L.K., A.K., and N.K. Protein expression and purification was conducted by L.K. and A.K. Cryo-EM grid preparation and screening were conducted by L.K. and A.K. Data processing, model building, and analysis were conducted by L.K. and A.K. L.K. conducted biophysical assays. Negative staining transmission electron microscopy grid preparation and screening was conducted by M.T.Ž. RT-qPCR and RNA sequencing was performed by L.K., T.K. and Ž.V. Mass spectrometry and mass photometry were conducted by J.L.P.B. and E.H. MD simulations were conducted by F.M. The manuscript was prepared by L.K. and M.P. All authors provided critical feedback and helped shape the research, analysis and manuscript.

## Competing interests

The authors declare no competing interests.
