## [Peer Review File · Communications Chemistry]

Reviewers' comments:

Reviewer #1 (Remarks to the Author):

Luka Kavčič and coworkers present an impressive manuscript focusing on the in-depth characterization of polymorphism in flexuous filamentous potato virus Y coat protein assemblies. Studying how viruses assemble and understanding the ways to direct these pathways, is an important and timely topic likely to attract attention from not only structural virologists, but also wider protein engineering community. Overall, the manuscript is clearly communicated and the detailed structural characterization has been carefully conducted. The presence of different control experiment is extensive and well-thought. I particularly enjoyed the fact that the work approaches its topic from various angles and aims not only to characterize the formed assemblies, but also their wider implications. Starting from the structural polymorphism of recombinant PVY CP and moving to characterizing the octameric rings and further mutations studies lay a strong basis for understanding the formation of cubic structures. Finally triggered assemblies, stabilization strategies and RNA packaging are studied.

My only detailed comment relates to the connection of the current manuscript to the work of Malay et al. (<https://doi.org/10.1038/s41586-019-1185-4>). The observation of protein rings forming cage structures and the role of disulfide bridges is similar in both works and it would make sense to point out this connection. Otherwise, (and very rarely), I'm happy to recommend the work for publication without further revisions.

Reviewer #2 (Remarks to the Author):

The paper provides a very thorough structural characterization of Potato virus Y (PVY) virions, in particular of virion-like particles (VLPs) formed by the polymorphic capsid protein (CP). CP is an incredibly plastic protein provided with intrinsically disordered N- and C-terminal domains and a partially conserved globular core that can be genetically engineered to self-assemble in a plethora of different VLPs conformations, most of which might have interesting nanobiotechnology applications. The aim of the study is to uncover most of these CP conformations to provide a large range of new nanoparticles with different physiochemical properties for vaccines, drug delivery systems, nanoreactors, biomaterials or nanomachines.

In this paper, the biochemical and structural characterization by negative stain EM and cryo-EM of every single VLP and trCP variant produced (14 and 17, respectively, as summarized in Supplementary Tables 1 and 2), as well as the level of detail in their analysis, including thermal stability assays, mass spectrometry and molecular dynamics simulations for some of the mutants, are remarkable. Almost every mutant has his own high-resolution 3D reconstruction from which is possible to infer more about key domain and residues positions and interactions and that is impressive. The paper is well written and, despite the large amount of data provided, it is very well structured and can be followed easily. I support the publication of this paper in Communications Chemistry journal because the results are novel and can be relevant to scientists in the specific sub-field of nanoparticles and nanobiotechnology in search for an alternative nanoparticle of interest.

Understanding that this is the first time this kind of PVY protein has been characterized and engineered and most of the results are novel and technically solid, I find that the only weak points of this study is the aim and possible applications of the newly discovered nanoparticles involving RNA. Overall, the authors claim they “can control the shape, size, RNA encapsidation, symmetry, stability and surface functionalization of nanoparticles through structure-based design of CP from potato virus Y (PVY)”. I agree with all the above claims, but I do not fully agree with the control over RNA encapsidation, yet. Experiments on VLPT43C+D136C filaments show that it encapsidates predominantly its own RNA and a few other random bacterial genes derived from the purification system of choice, while the reported attempts of heterologous p97 mRNA encapsidation were not efficient. The authors state that CPs from other potyviral families were already reported as capable of encapsidating heterologous viral RNA under certain conditions in vivo. Nonetheless, it seems that PVY CP has proven very limited specificity of RNA encapsidation so far, and if that cannot be strengthened and further investigated through focussed structure-based design of VLPs, it can narrow down the possible applications of PVY VLPs in nanobiotechnology.

Out of curiosity and in light of possible medical, biotechnological, or smart material applications, can the authors speculate a bit more on what is the future direction for these specific synthetic PVY VLPs they so carefully and thoroughly investigated, and propose what their best applications can be? As medical application, for example, how feasible is that some of these nanoparticles can be used for mRNA/protein delivery as an alternative to currently commercialized mRNA-lipid nanoparticle vaccines? How specialized their delivery can be on a specific receptor in human body/tissue? How can they open/unfold to release their cargo? How will they be degraded/cleared?

Minor comments:

- Title: the use of “metamorphosis” in the title could sound like a nice metaphore but the term has a specific meaning in zoology and does not describe what actually happens to the CP protein. I would suggest changing it and define it as mentioned in the main text “pleomorphism”, which accounts for the existence of irregular and variant forms in the same species or strain of microorganisms.
- Line 41: “generally encodes ten proteins³, which includes also the CP, which copies [...]” I suggest substituting with “whose copies [...]”.
- Line 284: “C2 symmetry axis” instead of “C2 axis”
- Line 533: “the conserved RNA binding loop S125-G130”. That loop is not showed as conserved in Supplementary Figure 7...it would be nice to highlight the residue conservation there too.
- Supplementary Figure 13c: VI in dark gray on the right should be VIII
- Supplementary Figure 18a, figure on the left: can the authors comment on the fibrillar assembly visible in the center?

Reviewer #3 (Remarks to the Author):

I co-reviewed this manuscript with one of the reviewers who provided the listed reports as part of a Communications Chemistry initiative to facilitate training in peer review and appropriate recognition for Early Career Researchers who co-review manuscripts.

RE: Point-by-point responses to reviewers' comments, COMMSCHEM-23-0480-T

Thank you very much for reviewing our manuscript COMMSCHEM-23-0480-T entitled “**From structural polymorphism to metamorphosis of the coat protein of flexuous filamentous potato virus Y**” by Kavčič *et al.*

Below you will find **our responses to the reviewers' comments point by point**. The reviewers' comments are in black font and our responses (A) are **in blue**. Further minor corrections have been explained to the Editor in the Cover letter. All changes in the main or supplementary text, figures or tables are highlighted in yellow.

Reviewer #1:

Luka Kavčič and coworkers present an impressive manuscript focusing on the in-depth characterization of polymorphism in flexuous filamentous potato virus Y coat protein assemblies. Studying how viruses assemble and understanding the ways to direct these pathways, is an important and timely topic likely to attract attention from not only structural virologists, but also wider protein engineering community. Overall, the manuscript is clearly communicated and the detailed structural characterization has been carefully conducted. The presence of different control experiment is extensive and well-thought. I particularly enjoyed the fact that the work approaches its topic from various angles and aims not only to characterize the formed assemblies, but also their wider implications. Starting from the structural polymorphism of recombinant PVY CP and moving to characterizing the octameric rings and further mutations studies lay a strong basis for understanding the formation of cubic structures. Finally triggered assemblies, stabilization strategies and RNA packaging are studied.

My only detailed comment relates to the connection of the current manuscript to the work of Malay et al. (<https://doi.org/10.1038/s41586-019-1185-4>). The observation of protein rings forming cage structures and the role of disulfide bridges is similar in both works and it would make sense to point out this connection. Otherwise, (and very rarely), I'm happy to recommend the work for publication without further revisions.

A. We are aware of this great work by Malay et al. 2019 and other related studies. However, as highlighted in our study, the unique feature of our cubes and spheres is that they are not held together by disulphide bridges or/and metal ions, which is in contrast to Malay et al. 2019, where the

assemblies are held together by S–Au–S staples between the protein oligomers. As explained in our manuscript, the Cys residues in trCP^{K176C} are too far apart to form disulphide bonds (Fig. 5a). Moreover, cubes can form even without Cys residues on the surface of the octameric rings (trCP^{K176S}, trCP^{K177E}) (Fig. 4, Supplementary Figs. 10c, 13b), and spherical particles are formed without Cys residues exposed on the surface of the octameric rings (trCP^{K177E}) (Fig. 4). trCP^{E150C}, forming cross-shaped and spherical junctions, also does not form disulphide bonds, as indicated by the SDS PAGE gel +/- DTT (Suppl. Fig. 9a), and the position of E150C on the N-side of the rings. This is because the N-sides of the stacked rings face away from the particle centre (Fig. 3d) and as such are not involved in the orthogonal interactions between the rings and double rings.

The only metal cluster we believe to be present in our structures is in double octameric rings (H2T or H2H) held together mainly by clustered His₆-tags in the centres of the rings (Fig. 2c, Supplementary Fig. 11c), which probably chelate Ni²⁺ as it can occasionally leak from the NiNTA resins. When these His₆-tags are removed by TEV protease, the double rings fall apart to form single octameric rings (Fig. 2c, Supplementary Fig. 11e). We can also see such His₆-tag clusters on the cube surfaces, which can be excised out by TEV protease. However, this does not lead to destabilization of the cubes (Fig. 5b). In addition, cubes can be assembled *in vitro* from the trCP^{K176C-noHis}-MBP fusion without His₆-tag on the trCP scaffold and therefore without Ni-His₆-tag chelation, but the cubes can still form.

Thus, we show that our cubic or spherical assemblies do not directly depend on disulphide bond and/or the metal ions, which is also explained in the text and figures.

However, we reference Sharma et al., 2022 (<https://doi.org/10.1021/acsnanoscienceau.2c00019>), line 519 in the discussion to refer to recombinant (or artificial) protein cages, in which (quote from Sharma et al) ‘a simple change in the position of a single amino acid responsible for Au(I)-mediated subunit–subunit interactions in the constituent ring-shaped building blocks results in a more acute dihedral angle between them’ and thus distinct final assembly (i.e., polymorphism).

Reviewer #2:

The paper provides a very thorough structural characterization of Potato virus Y (PVY) virions, in particular of virion-like particles (VLPs) formed by the polymorphic capsid protein (CP). CP is an incredibly plastic protein provided with intrinsically disordered N- and C-terminal domains and a partially conserved globular core that can be genetically engineered to self-assemble in a plethora of different VLPs conformations, most of which might have interesting nanobiotechnology applications. The aim of the study is to uncover most of these CP conformations to provide a large range of new nanoparticles with different physiochemical properties for vaccines, drug delivery systems, nanoreactors, biomaterials or nanomachines.

In this paper, the biochemical and structural characterization by negative stain EM and cryo-EM of every single VLP and trCP variant produced (14 and 17, respectively, as summarized in Supplementary Tables 1 and 2), as well as the level of detail in their analysis, including thermal stability assays, mass spectrometry and molecular dynamics simulations for some of the mutants, are remarkable. Almost every mutant has his own high-resolution 3D reconstruction from which is possible to infer more about key domain and residues positions and interactions and that is impressive. The paper is well written and, despite the large amount of data provided, it is very well structured and can be followed easily. I support the publication of this paper in Communications Chemistry journal

because the results are novel and can be relevant to scientists in the specific sub-field of nanoparticles and nanobiotechnology in search for an alternative nanoparticle of interest.

Understanding that this is the first time this kind of PVY protein has been characterized and engineered and most of the results are novel and technically solid, I find that the only weak points of this study is the aim and possible applications of the newly discovered nanoparticles involving RNA. Overall, the authors claim they “can control the shape, size, RNA encapsidation, symmetry, stability and surface functionalization of nanoparticles through structure-based design of CP from potato virus Y (PVY)”. I agree with all the above claims, but I do not fully agree with the control over RNA encapsidation, yet. Experiments on VLPT43C+D136C filaments show that it encapsidates predominantly its own RNA and a few other random bacterial genes derived from the purification system of choice, while the reported attempts of heterologous p97 mRNA encapsidation were not efficient. The authors state that CPs from other potyviral families were already reported as capable of encapsidating heterologous viral RNA under certain conditions *in vivo*. Nonetheless, it seems that PVY CP has proven very limited specificity of RNA encapsidation so far, and if that cannot be strengthened and further investigated through focussed structure-based design of VLPs, it can narrow down the possible applications of PVY VLPs in nanobiotechnology.

A. We thank the reviewer for this comment. We originally stated in the abstract that ‘we can control the shape, size, **RNA encapsidation**, symmetry, stability and surface functionalization of nanoparticles by structure-based design of CP from potato virus Y (PVY). By ‘controlling RNA encapsidation’ we meant, as explained later in the results section, that we can control whether VLPs formed by PVY CP derivatives can contain RNA or not. This is an important result, as for certain potential applications, one does not want to introduce foreign RNA into the system. Here we offer customized VLPs that can be produced in the complete absence of RNA. Or, if we want more homogeneous particles (in terms of dimensions) and are not concerned about the presence of RNA, we can produce only RNA-encapsidating VLPs, such as VLP^{T43C+D136C}. We agree with the reviewer and point out in the manuscript that we cannot (yet) produce VLPs that contain only one specific type of RNA, as our results have shown that recombinant PVY CP has limited specificity and can therefore bind different RNAs (but as shown here, not just any). Our work thus lays the groundwork for future studies that will help us understand if and how we can achieve specificity for RNA packaging by PVY CP *in vitro*. If discovered, such VLPs could be used as an RNA storage system, as the capsid protein protects the RNA packaged within, or as a potential RNA delivery system.

Therefore, in our manuscript, a wide array of quaternary structures formed by CP (and its derivatives) is shown, also showing potential for further modifications and possible applications. To avoid further confusion with regards to RNA packaging, we have slightly changed the abstract and added the word ‘ability’ to RNA encapsidation: ‘Here, we show that we can control the shape, size, RNA encapsidation **ability**, symmetry, stability and surface functionalization of nanoparticles through structure-based design of CP from potato virus Y (PVY).’

Out of curiosity and in light of possible medical, biotechnological, or smart material applications, can the authors speculate a bit more on what is the future direction for these specific synthetic PVY VLPs they so carefully and thoroughly investigated, and propose what their best applications can be?

A. Different VLP (filament) architectures (stacked rings, helical) offer alternative (symmetric) surface patterns that could be further explored for their use in applications. Depending on the filament surface modification (fusions of molecules on the outer surface or in the lumen), one could envision use of engineered PVY VLPs in vaccine development, biosensors, delivery systems ([10.1016/j.addr.2018.08.011](https://doi.org/10.1016/j.addr.2018.08.011)) or as organic scaffolds in catalysis ([10.1039/C5CS00287G](https://doi.org/10.1039/C5CS00287G)). Furthermore, our results have provided ways to optimize their biophysical stability, with disulphide bond-stapled PVY VLPs (with T_m values between 60 and 70 °C) achieving comparable stability to nanoparticles based on the PVX virus (filamentous flexible potato virus X from the potexvirus genus) ([10.1002/1873-3468.12184](https://doi.org/10.1002/1873-3468.12184)). We speculate that additional stabilization could be achieved by further protein engineering of CP-CP interaction sites as well as by introducing additional staples to further stabilize intramolecular CP interactions.

We have also shown that trCP assemblies have a range of shapes and sizes. The most attractive appears to be the cubic assembly, which offers a small octahedral scaffold (16.5 nm diameter, 48 copies of the same protein, homogeneous cubic particles) with the potential for specific surface modification (via His-tag or SpyTag attached to each of 48 copies) in the area of multivalent vaccine development or cargo delivery. The possibility of *in vitro* assembly could be employed in future to capture biomolecules within filaments or cubes/spheres thus serving also as potential (delivery) protein cage.

As medical application, for example, how feasible is that some of these nanoparticles can be used for mRNA/protein delivery as an alternative to currently commercialized mRNA-lipid nanoparticle vaccines?

A. In contrast to the already commercialized mRNA-lipid (spherical) nanoparticle vaccines (for specific applications) (<https://www.nature.com/articles/s41578-021-00358-0>), the applications of flexuous filamentous capsids are, to our knowledge, not yet on the market. However, there are several studies that show how these elongated flexuous particles, which have a large surface-to-volume ratio and thus offer a high payload capacity and an advantage over the icosahedral viruses, could be used in medicine for various purposes, such as antigen presentation, antibody sensing, enzyme nanocarriers, drug delivery, immunodiagnostics, high-resolution imaging, etc (DOI: 10.1016/bs.aivir.2020.09.001). Interestingly, the elongated morphology offers increased tumour homing and tissue penetration compared to spherical particles ([10.1021/mp300240m](https://doi.org/10.1021/mp300240m)). The main mode of internalization of VLPs from the potyvirus Pepper vein banding virus (PVBV) was shown to be caveolae-mediated endocytosis in both HeLa and HepG2 cells via cell surface proteins (10.2217/nmm-2018-0405). Thus, potyvirus VLPs use the same attachment/entry mechanisms into mammalian cells as animal picornaviruses, with PVBV VLPs found to be degraded in the lysosomes. Furthermore, due to the filamentous helical packing of RNA, potyviral capsids should in theory impose no limitations on the size of encapsidated RNA cargo and could be used for the storage and/or delivery of mRNAs of different lengths (from short to very long), once it is known how to achieve the specificity. While the entrapment of target RNA in lipid nanoparticles is mainly non-selective, some filamentous capsids such as TMV can achieve efficient encapsidation via recognition of specific RNA motifs known as

origin of assembly sequences ([10.1016/0092-8674\(77\)90065-4](https://doi.org/10.1016/0092-8674(77)90065-4), [10.1016/j.virol.2005.01.018](https://doi.org/10.1016/j.virol.2005.01.018)), decreasing the chance of additional undesirable cargo.

To actually utilize these theoretical advantages of flexuous potyviral VLPs, future investigation into virion assembly/RNA encapsidation process are required for efficient encapsidation of desired mRNA into potyviral (in our case PVY) VLPs either *in vitro* or in an expression system such as bacteria. As mentioned above, our study could be an excellent starting point for future studies to understand whether and how one could achieve packaging of specific RNA sequences by PVY CP in a recombinant environment, which would further pave the way for their use in the biomedical field.

How specialized their delivery can be on a specific receptor in human body/tissue? How can they open/unfold to release their cargo? How will they be degraded/cleared?

A. These questions were to some extent already addressed above. Furthermore, the specificity of cargo delivery via VLP scaffolds could be achieved based on nanoparticle surface functionalization via various targeting strategies (external surface attachment of small molecules or specific proteins as receptor ligands or antibodies recognizing specific human receptor on the target cell ([10.1016/j.cell.2020.02.001](https://doi.org/10.1016/j.cell.2020.02.001))). On the other hand, due to their high aspect ratio, PVX virions as flexuous filamentous nanoparticles already showed enhanced tumour homing and tissue penetration properties in mice compared to spherical CPMV particles ([10.1021/mp300240m](https://doi.org/10.1021/mp300240m)).

In the case of purely proteinaceous VLPs, biodegradability should not be a problem (proteolysis). One would also expect relatively similar biodegradability with RNA encapsidating VLPs (proteases, nucleases). Of course, future studies are required to determine their bioavailability, clearance and cargo release *in vivo*.

One of the main goals of our work was to show in detail a wide spectrum of structural possibilities (i.e. atomic models of scaffolds and their properties that can be further modified) offered by PVY CP, which in our opinion provide attractive directions for future studies on potyviral VLPs. To expose the potential of nanoparticles shown here, we have added a line to the last paragraph of discussion, lines 577-578: The high-resolution data obtained in this study and the possibility of structure-based design of nanoparticles with novel architectures and tailored properties make PVY CP an excellent candidate for nanobiotechnological applications, **such as vaccine and biosensor development, cargo storage and delivery, medical imaging, or energy and nanostructured materials.**

Minor comments from the reviewer #2:

• Title: the use of “metamorphosis” in the title could sound like a nice metaphor but the term has a specific meaning in zoology and does not describe what actually happens to the CP protein. I would suggest changing it and define it as mentioned in the main text “pleomorphism”, which accounts for the existence of irregular and variant forms in the same species or strain of microorganisms.

A. We thank the reviewer for this comment. Indeed, we have also discussed and read various definitions of polymorphism, pleomorphism and metamorphosis, that helped us to define the title.

Structural **polymorphism**: ‘The functions of biological macromolecules are often associated with conformational malleability of the structures. This phenomenon of chemically identical molecules with different structures is coined structural polymorphism.’ ([10.3389/fbinf.2021.788308](https://doi.org/10.3389/fbinf.2021.788308))

Virus **pleomorphism** can be defined by the irregularity of particles from the same virus species, i.e., the variation of size and shape as well as the protein copy number that constitute the virus particle. In this case, pleomorphism thus describes existence of different structural states of the virus, which is natural polymorphism in live viruses (DOI: 10.1039/d3tb00991b).

In our case, the term ‘structural polymorphism’ was used to show how a single type of CP (either wild-type, CP of full length, or CP^{ΔC40} etc, or trCP^{E150C}) can simultaneously form VLPs of different architectures. We believe that in our case (at the molecular/structural level), using the term pleomorphism would mean the same as polymorphism.

‘Metamorphosis is a biological process by which an animal physically develops including birth transformation or hatching, involving a conspicuous and relatively abrupt change in the animal's body structure through cell growth and differentiation’ (Wikipedia)

At the molecular level, metamorphosis could represent (strikingly) different architectures by a single type of protein under certain conditions (transition from one form to another). The term molecular metamorphosis has also been used to describe the changes in the quaternary structure of proteins (<https://doi.org/10.1111/j.1742-4658.2010.07671.x>), and can also be observed in polymers (<https://www.chemistryworld.com/news/polymers-undergo-molecular-metamorphosis/2500433.article>). In our case, ‘metamorphosis’ was used to describe the transformation-metamorphosis of monomers into filamentous or cubic structures, of filaments into rings, and from rings back to filaments or cubes or spheres or junctions of filaments (using mutations, deletions, assembly in bacteria, *in vitro* etc). We therefore believe that the use of the term ‘(structural) metamorphosis’ is appropriate, and in our opinion, should remain in the title. To substantiate this, we have made a small change in the discussion, in lines 562-563, ‘Fifth, we show that we can achieve a striking change in quaternary structure, i.e. molecular metamorphosis, by simple genetic modifications of CP. By deletions and/or single-site mutations, we can reduce or even prevent the filament formation and instead produce single or double octameric rings of CP as well as highly ordered cubic or spherical self-assemblies of these rings, which can be further modified to form into cross shaped forms (Figs. 3-5)’.

• Line 41: “generally encodes ten proteins³, which includes also the CP, which copies [...]” I suggest substituting with “whose copies [...]”.

A. Thank you for this suggestion, it has been corrected as suggested.

• Line 284: “C2 symmetry axis” instead of “C2 axis”

A. Thank you, it has been corrected as suggested.

- Line 533: “the conserved RNA binding loop S125-G130”.

That loop is not showed as conserved in Supplementary Figure 7...it would be nice to highlight the residue conservation there too.

A. Thank you, it has been added as suggested in the Supplementary Figure 7.

- Supplementary Figure 13c: VI in dark gray on the right should be VIII

A. Thank you very much for noting the typo, it has been corrected in the Supplementary Figure 13c, as suggested.

- Supplementary Figure 18a, figure on the left: can the authors comment on the fibrillar assembly visible in the center?

A. Thank you for this comment. The fibrillar aggregate in the centre of Supplementary Fig. 18a, is structurally distinct from properly assembled VLPs (Supplementary Fig. 18c). It could result from a presence of the free CP in the sample before further purification (see SEC (panel b) fraction A and its SDS-PAGE in panel c), as we can see a significant band for CP in the fraction A (no TEV protease present – probably leaky cleavage). This CP could associate with itself or CP-MBP, or some other impurity. However, it does not seem to form VLPs of a regular structure. These longer aggregates (fraction A in SEC) are removed prior triggering *in vitro* assembly from the monomeric CP-MBP units. Namely, the sample imaged by negative staining TEM shown on Supplementary Fig. 18a (left) was purified further using SEC (middle panel). According to the elution volume, the fraction B should contain CP-MBP monomers (free CP is negligible on SDS-PAGE for CP-MBP (fraction B)), and neg. stain TEM proved that there were no elongated aggregates in the fraction B (Supplementary Fig. 18a, micrograph on the right). As further shown in the figure, this fraction B was then then treated by the TEV protease, thereby releasing CP, which triggered formation of properly structured VLPs (Supplementary Fig. 18c), which has been also described in the figure legend.

Reviewer #3:

I co-reviewed this manuscript with one of the reviewers who provided the listed reports as part of a Communications Chemistry initiative to facilitate training in peer review and appropriate recognition for Early Career Researchers who co-review manuscripts.

Sincerely yours,

Assoc. Prof. Marjetka Podobnik, corresponding author

On behalf of the authors

REVIEWERS' COMMENTS:

Reviewer #2 (Remarks to the Author):

Regarding the authors' response to the first minor comment:

The New Oxford American Dictionary defines 'metamorphosis' as:

1. zoology (in an insect or amphibian): the process of transformation from an immature form to an adult form in two or more distinct stages;
2. a change of the form or nature of a thing or person into a completely different one.

Following the citations kindly provided by the authors above and researching also about the 'molecular metamorphosis' concept I found: "The molecular metamorphosis² defines the situation of a molecule (generator) capable by external inputs of giving a different molecule and another one successively, leading to a set of new molecules", as defined in PMID 34805653. I believe that this definition does not apply to this paper because no external input is given (e.g., temperature change) nor a set of new molecules is created, rather a new set of molecular arrangements/architectures from the same building unit, the CP protein. For this reason, I agree with the definition 'structural metamorphosis' as used in PMID 34665484 and PMID 36891540 to describe accurately the plasticity and quaternary structure heterogeneity reported in this article for PVY CP protein. Therefore, I would again suggest adapting the title to something like: "From pleomorphism to structural metamorphosis of the flexuous potato virus Y coat protein", 14 words in total.

Apart from this, I have no further remarks. I compliment the authors for the remarkable work presented and thank them for the thorough responses to all my questions and comments. I am happy to support the publication of this paper in Communication Chemistry because of the high quality and relevance of scientific results provided within.

Reviewer #3 (Remarks to the Author):

I co-reviewed this manuscript with one of the reviewers who provided the listed reports as part of a Communications Chemistry initiative to facilitate training in peer review and appropriate recognition for Early Career Researchers who co-review manuscripts.

RE: Point-by-point responses to reviewers' comments, COMMSCHEM-23-0480A

Thank you for a further revision of our manuscript COMMSCHEM-23-0480A originally entitled “**From structural polymorphism to metamorphosis of the coat protein of flexuous filamentous potato virus Y**” by Kavčič *et al.*

Below you will find **our responses to the reviewers' comments point by point**. The reviewers' comments are shown in black and our responses (A) in blue.

Reviewer #2:

Regarding the authors' response to the first minor comment:

The New Oxford American Dictionary defines 'metamorphosis' as:

1. zoology (in an insect or amphibian): the process of transformation from an immature form to an adult form in two or more distinct stages;
2. a change of the form or nature of a thing or person into a completely different one.

Following the citations kindly provided by the authors above and researching also about the 'molecular metamorphosis' concept I found: "The molecular metamorphosis² defines the situation of a molecule (generator) capable by external inputs of giving a different molecule and another one successively, leading to a set of new molecules", as defined in PMID 34805653. I believe that this definition does not apply to this paper because no external input is given (e.g., temperature change) nor a set of new molecules is created, rather a new set of molecular arrangements/architectures from the same building unit, the CP protein. For this reason, I agree with the definition 'structural metamorphosis' as used in PMID 34665484 and PMID 36891540 to describe accurately the plasticity and quaternary structure heterogeneity reported in this article for PVY CP protein. Therefore, I would again suggest adapting the title to something like: "From pleomorphism to structural metamorphosis of the flexuous potato virus Y coat protein", 14 words in total.

Apart from this, I have no further remarks. I compliment the authors for the remarkable work presented and thank them for the thorough responses to all my questions and comments. I am happy to support the publication of this paper in Communication Chemistry because of the high quality and relevance of scientific results provided within.

A. We thank the reviewer for this additional and kind contribution. We agree that we should emphasize 'structural' metamorphosis (rather than 'molecular' metamorphosis or just 'metamorphosis'). Therefore, we have corrected line 425? in Discussion, and replaced 'molecular metamorphosis' with 'structural metamorphosis'.

We also thank you for suggesting the new title of the manuscript. However, we would like to stick to the use of 'polymorphism' instead of 'pleomorphism', the latter word was suggested by the reviewer. In most of the papers in the field of virus-like particles or nanoparticles that we have come across (for example Lie et al, 2023, DOI: 10.1039/d3tb00991b, Obr & Schur, 2019 <https://doi.org/10.1016/bs.aivir.2019.07.008>, Wang et al. 2021, <https://doi.org/10.1038/s41467-020-20689-w>, Wang et al, 2022 <https://doi.org/10.1021/acs.chemrev.1c00753>) as well as according to Oxford Languages definition, 'pleomorphism' should represent the occurrence of more than one distinct form of a natural object, such as a crystalline substance, virus, the cells in a tumor, or an organism at different stages of the life cycle. In line with aforementioned publications, we believe that the term 'structural polymorphism' (the formulation we have used so far) is more appropriate, as we believe it has a broader meaning (and in line with its use in the literature) and includes *in vitro* produced nanoparticles in addition to natural objects (as described in our manuscript).

Secondly, we also emphasize in our title, that the coat protein originates from the flexuous filamentous virus (to our knowledge, this is the first such study of on type of virus).

We therefore suggest the following for our title: 'From structural polymorphism to structural metamorphosis of the coat protein of flexuous filamentous potato virus Y'.

The word 'structural' may seem redundant in this case at first sight, but on the other hand we emphasize that in both cases we are talking about a 'structural' phenomenon. Of course, we are open to further suggestions, as native speakers may know/understand better than we do how to connect the adjective 'structural' to both 'polymorphism' and 'metamorphosis' without its repetition.

Reviewer #3:

I co-reviewed this manuscript with one of the reviewers who provided the listed reports as part of a Communications Chemistry initiative to facilitate training in peer review and appropriate recognition for Early Career Researchers who co-review manuscripts.

Sincerely yours,

Assoc. Prof. Marjetka Podobnik, corresponding author

On behalf of the authors